# Right Now, Wrong Then: Non-Stationary Direct Preference Optimization under Preference Drift

## Abstract

Current Large Language Model (LLM) preference optimization algorithms do not account for temporal preference drift, which can lead to severe misalignment. To address this limitation, we propose an offline fine-tuning algorithm *Non-Stationary Direct Preference Optimisation* (NS-DPO) which models time-dependent reward functions with a Dynamic Bradley-Terry model. NS-DPO applies exponential weighting, by introducing a discount parameter in the loss function, which proportionally focuses learning on more time-relevant datapoints. We theoretically analyse the convergence of NS-DPO, providing upper bounds on the estimation error and regret caused by non-stationary preferences. Finally, we demonstrate the effectiveness of NS-DPO[1] for fine-tuning LLMs in scenarios with drifting preferences. By simulating preference drift using popular LLM reward models and datasets accordingly, we show that NS-DPO fine-tuned LLMs remain robust under non-stationarity, significantly outperforming baseline algorithms that ignore temporal preference changes, without sacrificing performance in stationary cases.

## 1 Introduction

The application of Reinforcement Learning from Human Feedback (RLHF) to fine-tune Large Language Models (LLMs) (Christiano et al., 2017; Stiennon et al., 2020; Ziegler et al., 2019; Ouyang et al., 2022; Bai et al., 2022b) has lead to more precise control over the behaviour they exhibit. This control is crucial when looking to safely deploy models in the real world (Amodei et al., 2016; Hendrycks & Mazeika, 2022). Human preference datasets enable the training of proxy *reward models* (see, e.g., RewardBench (Lambert et al., 2024)) that can accurately evaluate complex human behaviour. These proxy reward models are used in conjunction with RL to fine-tune the LLM. Recent works (Rafailov et al., 2024; Azar et al., 2024; Hong et al., 2024) seek to improve the efficiency and stability of these approaches (Chaudhari et al., 2024) by training the LLM straight from human preference data, avoiding the need to learn a proxy reward model.

A key assumption made in these preference optimization algorithms is that human preferences are *stationary*, i.e., they do not change over time. However, a sudden or gradual shift in preferences can occur due to new information becoming available (Zafari et al., 2019; Johnson & Mayorga, 2020), changes in the demographics of the queried audience (Caldwell, 1981), or social influences and cultural trends. As more preference datasets are gathered over long periods of time, the chance of the data containing varying preferences increases. In such cases, algorithms that do not account for these changes, view them as noise and treat outdated data as equally important as fresh data, often leading to deteriorated performance. An increasing body of evidence (Zhou et al., 2024; Chen et al., 2024a) points to data quality as being a key factor in fine-tuning performance, thus preference drift can greatly affect the alignment of models which do not account for it (Carroll et al., 2024). The development of preference optimization algorithms and theory to handle preference drifts are therefore crucial.

In this work, we propose *Non-Stationary Direct Preference Optimization* (NS-DPO), a novel approach that uses a probabilistic *Dynamic* Bradley-Terry model (Cattelan et al., 2013; Bong et al., 2020; Tian et al., 2023) to account for non-stationary drift in human preferences. NS-DPO

---

[1]For code, see https://anonymous.4open.science/r/ns-dpo-CD67/.

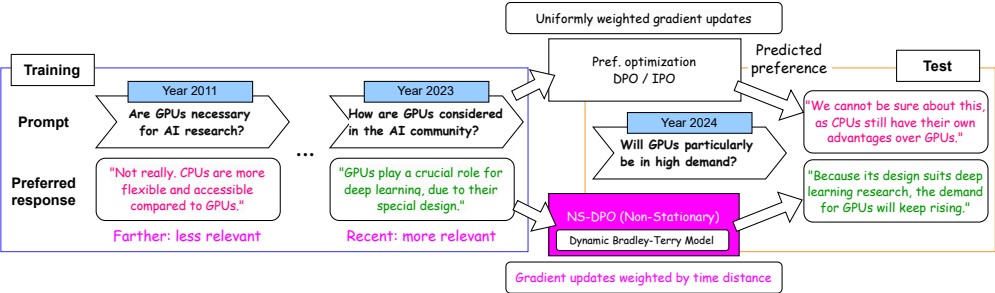

Figure 1: Human preferences are dynamic and influenced by a variety of factors (e.g. environment change and societal influence). However, standard preference optimization approaches (e.g., DPO and IPO (Rafailov et al., 2024; Azar et al., 2024)) do not account for this non-stationarity. In contrast, NS-DPO robustly learns on non-stationary data by using a **Dynamic Bradley-Terry model**, and adjusts the loss to discount older datapoints and concentrate learning on the latest data.

re-weights each training datapoint by appropriately down-weighting older data with potentially stale preferences and up-weighting more recent ones. We empirically show the effectiveness and robustness of NS-DPO compared to stationary approaches, using both synthetic experiments and datasets commonly used for fine-tuning LLMs. Our overall approach is summarised in Figure 1.

**Related work.** One of the primary applications of the RLHF framework is fine-tuning large language models (LLMs) (Christiano et al., 2017; Stiennon et al., 2020; Ziegler et al., 2019; Ouyang et al., 2022; Bai et al., 2022b). A key component of this is the Bradley-Terry model (Bradley & Terry, 1952) which learns a reward signal from paired human preferences. Rafailov et al. (2024) propose Direct Preference Optimization (DPO), which implicitly uses the Bradley-Terry model, to fine-tune an LLM directly from a preference dataset. A variety of alternatives to DPO have been proposed which adapt or do not use the Bradley-Terry model (Azar et al., 2024; Amini et al., 2024; Meng et al., 2024; Cen et al., 2024; Xu et al., 2023). Ethayarajh et al. (2024) remove paired preferences and propose maximising a utility function. Our work is the first to consider a direct preference algorithm using a Dynamic Bradley-Terry model.

A variety of work has also analysed the RLHF problem from a theoretical standpoint. Xiong et al. (2024) provide suboptimiality bounds of policies in the offline, online and hybrid settings under linear rewards. They do not directly analyse the performance of DPO, but propose it as a practical implementation of the oracle central to their analysis. Zhu et al. (2023); Chowdhury et al. (2024) analyse the offline preference learning and DPO settings, respectively. Chowdhury et al. (2024) address noisy preferences with a modified version of the DPO algorithm, presenting confidence bounds for neural policy classes and suboptimality bounds for the setting with log-linear policies.

Parameter drift has been widely studied in the bandit literature. Cheung et al. (2019) propose using a sliding window to estimate parameters with data points close to the current timestep, whilst Bogunovic et al. (2016); Zhao et al. (2020) investigate a restarting strategy. Similarly to the strategy of Russac et al. (2019), we use an exponentially weighted discounting term to re-weight points close to the current timestep. Faury et al. (2021); Wang et al. (2023) apply this approach to the case of generalised linear bandits first proposed by Filippi et al. (2010). Pacchiano et al. (2021); Saha (2021); Mehta et al. (2023) focus on the duelling bandit setting, where only preference feedback between two actions is provided by the environment. In this work, we provide the *first* theoretical guarantees for the popular offline setting where the true reward parameter (used to label training data) is allowed to change over time.

**Main contributions.** We propose NS-DPO, a direct preference optimization method that accounts for non-stationary preferences in the dataset via a Dynamic Bradley-Terry model. NS-DPO modifies the training loss with a single exponential weighting parameter $\gamma$, and thus represents a simple and practical extension of the popular DPO algorithm. We provide an upper bound on the regret of NS-DPO for log-linear policies given standard data coverage assumptions used in offline learning. To explore the performance of NS-DPO, we construct *non-stationary preference datasets* from a variety of existing popular datasets; including GlobalOpinionsQA (Durmus et al., 2023), Helpful & Harmless (Dai et al., 2024), and UltraFeedback (Cui et al., 2023). We demonstrate that NS-DPO significantly outperforms stationary DPO and other relevant baselines on these non-stationary datasets with varying degrees of preference drift on Llama models Touvron et al. (2023); Dubey et al. (2024).

## 2  PRELIMINARIES

**Stationary RLHF.** In the stationary RLHF setting (Ziegler et al., 2019; Ouyang et al., 2022), the goal is to find a suitable LLM policy $\pi$, whose response $a$, to a prompt $x$, maximise a reward function $r(x, a)$, i.e.,

$$\mathcal{J}(\pi) = \mathbb{E}_{x \sim \mathcal{X}, a \sim \pi}\left[r(x, a) - \tau D_{\mathrm{KL}}[\pi(\cdot|x)\|\pi_{\mathrm{ref}}(\cdot|x)]\right]. \tag{1}$$

Here, the KL-divergence prevents the learnt policy from deviating too far from some reference policy $\pi_{\mathrm{ref}}$, that has characteristics we wish to preserve in the final model. This is controlled by the parameter $\tau > 0$. In practical settings, human feedback is too complex to capture in a hand designed reward model, and we resort to learning a model from human preference data.

**Bradley-Terry Model.** A human preference dataset consists of prompts and two possible responses $\mathcal{D} = \{(x_i, a_i, a_i')\}_{i \in [n]}$, where $a_i$ is the response preferred to $a_i'$, and $n$ is the number of datapoints. To learn a reward model from this dataset, we assume the preferences are generated by a Bradley-Terry (BT) model (Bradley & Terry, 1952) where the probability that $a_i$ is preferred to $a_i'$ is

$$p(a_i \succ a_i'|x_i) = \sigma(r(x_i, a_i) - r(x_i, a_i')). \tag{2}$$

In Equation (2), $\sigma(\cdot)$ is the logistic sigmoid function and $r(x, a)$ is the reward model of human preferences we do not have access to and wish to learn. We parameterise the reward, typically as a single layer MLP on the last layer of the reference policy model $\pi_{\mathrm{ref}}$ (Ziegler et al., 2019), and then learn the parameters using a maximum likelihood estimator. An LLM can then be fine-tuned on the objective in Equation (1) using Reinforcement Learning (RL). It is important to note that the BT model captures many of the inherent assumptions we make about our data, which include the stationary nature of the underlying data generating process.

**Direct Preference Optimization.** Recent work by (Rafailov et al., 2024) avoids the training of an explicit reward model in the stationary RLHF process by optimizing the LLM policy directly from human preference data. To do this, the analytical solution to the stationary RLHF objective is rearranged into Equation (1) to derive an implicit reward

$$r(x, a) = \tau \log \frac{\pi(a|x)}{\pi_{\mathrm{ref}}(a|x)} + \tau \log Z(x), \tag{3}$$

where $Z(x)$ is a normalisation constant. This is substituted into the negative log likelihood of the Bradley-Terry model (see Equation (2)) resulting in the direct preference optimization (DPO) objective

$$\mathcal{L}(\pi) = \sum_{(x, a, a') \in \mathcal{D}} - \log \sigma\left(\tau \log \frac{\pi(a|x)}{\pi_{\mathrm{ref}}(a|x)} - \tau \log \frac{\pi(a'|x)}{\pi_{\mathrm{ref}}(a'|x)}\right). \tag{4}$$

All the methods introduced in this section, including DPO, are all stationary as they assume the reward model does not change with time. However, this assumption does not hold when training on real-world data. The changes in preferences over time, captured in the dataset, appear as label noise to the stationary methods.

## 3  LEARNING UNDER PREFERENCE DRIFT

To address the problem of preference drift, in datasets collected over a period of time, we propose *Non-Stationary Direct Preference Optimization* (NS-DPO). NS-DPO incorporates the *Dynamic Bradley-Terry* model, which includes a non-stationary reward model $r(x, a, t)$. Here $t \in \{1, \ldots, T-1\}$ denotes a time step in the past, and $T \in \mathbb{N}_+$ denotes the *current time step*, where we are evaluating the trained policy. Under the Dynamic Bradley-Terry model, the probability of response $a_i$ being preferred to $a_i'$ is

$$p(a_i \succ a_i'|x_i, t_i) = \sigma(r(x_i, a_i, t_i) - r(x_i, a_i', t_i)), \tag{5}$$

where in addition to the prompts and responses, we assume the dataset has temporal information about when the human preference between the two responses is expressed, $\mathcal{D} = \{(x_i, a_i, a_i', t_i\}_{i \in [n]}$. For the ease of indexing datapoints, we assume $t_i \leq t_j$ if $i < j$.

Rather than making an explicit assumption on how the reward function varies over time, we consider a setting in which the degree the reward can change is upper bounded. This is a mild assumption

on the temporal variation, and allows the reward to vary drastically at any point in time over all $T - 1$ steps over which our training data is recorded. We formalise this in Assumption 3 (Section 4), and use it to show that the convergence of NS-DPO depends upon the upper bound of the allowed drift. An approach to learning in this setting is via an *exponentially weighted maximum likelihood estimator* (Faury et al., 2021; Russac et al., 2019; Wang et al., 2023), where the datapoints are re-weighted such that losses incurred at the most recent datapoints are prioritised.

To learn a suitable reward model in this setting, we define the reward at time step $T$ as $r(x, a, T) \in \mathcal{R}$, where $\mathcal{R}$ is the space of reward values. We estimate the reward function at timestep $T$, by maximising the exponentially weighted negative log-likelihood of the Dynamic Bradley-Terry model:

$$\mathcal{L}_{DBT}(r) = \sum_{(x_i, a_i, a_i', t_i) \in \mathcal{D}} -\gamma^{T - t_i - 1} \log \sigma \left( r(x_i, a_i, T) - r(x_i, a_i', T) \right). \tag{6}$$

In Equation (6), $\gamma \in (0, 1)$ controls the rate at which older datapoints are discounted. The loss recovers the stationary Bradley-Terry model as $\gamma \to 1$.

**Offline Non-Stationary Direct Preference Optimization.** The derivation of NS-DPO follows as previously shown in Section 2 for the stationary case. We first define the RLHF objective at timestep $T$ as

$$\mathcal{J}_T(\pi) = \mathbb{E}_{x \sim \mathcal{X}, a \sim \pi} \left[ r(x, a, T) - \tau D_{KL}[\pi(\cdot|x) \| \pi_{\text{ref}}(\cdot|x)] \right], \tag{7}$$

where we are interested in maximising the reward function $r(x, a, T)$ that reflects human preferences in the present (i.e., the current time step). We note the prompt distribution $\mathcal{X}$ and the reference model $\pi_{\text{ref}}$ do not vary with time. As we consider the reward model at $T$, we derive an implicit reward of the same form as Equation (3). This relates the optimal policy and reward function of Equation (7) as

$$r(x, a, T) = \tau \log \frac{\pi_T^*(a|x)}{\pi_{\text{ref}}(a|x)} + \tau \log Z_T^*(x), \tag{8}$$

where $\pi_T^*$ is the optimal policy that optimises Equation (7) and $Z_T^*$ denotes the normalisation constant of $\pi_T^*$. We then parameterise the policy $\pi$ in Equation (7) using the parameter $\theta_T$, which enables expressing the implicit reward with respect to the parameter as

$$r_{\theta_T}(x, a, T) = \tau \log \frac{\pi_{\theta_T}(a|x)}{\pi_{\text{ref}}(a|x)} + \tau \log Z_{\theta_T}(x), \tag{9}$$

where $Z_{\theta_T}$ denotes the normalisation constant of $\pi_{\theta_T}$. We apply Equation (9) into the exponentially weighted negative log likelihood in Equation (6) to derive the NS-DPO objective

$$\mathcal{L}^{\text{NS}}(\theta_T) = \sum_{(x_i, a_i, a_i', t_i) \in \mathcal{D}} -\gamma^{T - t_i - 1} \log \sigma \left( \tau \log \frac{\pi_{\theta_T}(a_i|x_i)}{\pi_{\text{ref}}(a_i|x_i)} - \tau \log \frac{\pi_{\theta_T}(a_i'|x_i)}{\pi_{\text{ref}}(a_i'|x_i)} \right). \tag{10}$$

## 4 THEORETICAL ANALYSIS OF OFFLINE NON-STATIONARY DPO

In this section, we analyse the performance of NS-DPO in the offline setting. We assume the use of log-linear policies, and present how the preference drift affects the estimation error and regret bound of the algorithm. We provide the sample complexity of the algorithm, which recovers $O(n^{-1/2})$ when the preferences are stationary. See Appendix E for further details.

**Policy Class.** We use the policies parameterised by $\theta \in \Theta \subset \mathbb{R}^d$ of the following form

$$\Pi = \left\{ \pi_\theta(a|x) = \frac{\exp(f_\theta(x, a))}{\sum_{a' \in \mathcal{A}} \exp(f_\theta(x, a'))} \right\}, \tag{11}$$

where $f_\theta(x, a) \in \mathbb{R}$ is a differentiable function. For our analysis, we consider the case of log-linear policies where $f_\theta$ is linear: $f_\theta(x, a) = \phi(x, a)^\intercal \theta$, and the feature map $\phi(x, a)$ is a $d$-dimensional vector. This is motivated by the reward model introduced in Ziegler et al. (2019) where the last hidden layer of the LLM is used as the feature embedding function $\phi(x, a)$.

**Loss Function with $\ell_2$ regulariser.** For the analysis of log-linear policies, we regularise the NS-DPO loss with squared $\ell_2$-norm of $\theta$, $\tau^2$ and a non-linearity coefficient $c_{\sigma, \tau}$ (explained in Appendix E):

$$\mathcal{L}_{\text{reg}}^{\text{NS}}(\theta) = \frac{1}{n} \mathcal{L}^{\text{NS}}(\theta) + \frac{\lambda c_{\sigma, \tau} \tau^2}{2} \|\theta\|^2. \tag{12}$$

**Performance measure and Optimal Policy.** Let $\tilde{\theta}_T \in \Theta$ denote the parameter that minimises the (regularised) NS-DPO loss defined in Equation (12). We assess the performance of the policy $\pi_{\tilde{\theta}_T}$, using the difference of non-stationary RLHF objectives between $\pi_{\tilde{\theta}_T}$ and $\pi_T^*$ in Equation (7):

$$R_T^{\text{off}} = \mathcal{J}_T(\pi_T^*) - \mathcal{J}_T(\pi_{\tilde{\theta}_T})$$

$$= \mathbb{E}_{x \sim \mathcal{X}} \Big[ \mathbb{E}_{a \sim \pi_T^*(\cdot|x)}[r(x,a,T)] - \tau \mathrm{D}_{\mathrm{KL}}[\pi_T^*(\cdot|x)\|\pi_{\mathrm{ref}}(\cdot|x)]$$

$$- \mathbb{E}_{a' \sim \pi_{\tilde{\theta}_T}(\cdot|x)}[r(x,a',T)] + \tau \mathrm{D}_{\mathrm{KL}}[\pi_{\tilde{\theta}_T}(\cdot|x)\|\pi_{\mathrm{ref}}(\cdot|x)] \Big]. \tag{13}$$

Here $r(\cdot, \cdot, T)$ denotes the true reward function at time $T$, and $\pi_T^*$ denotes the optimal policy against which we compare the performance of our algorithm. Given a reference policy $\pi_{\mathrm{ref}}$, the optimal policy is defined as the policy which optimises the RLHF objective at time step $T$

$$\pi_T^* = \arg\max_{\pi \in \Pi} \mathbb{E}_{x \sim \mathcal{X}, a \sim \pi} \Big[ r(x,a,T) - \tau \mathrm{D}_{\mathrm{KL}}[\pi(\cdot|x)\|\pi_{\mathrm{ref}}(\cdot|x)] \Big]. \tag{14}$$

Similarly, we can define the parameter $\theta_t^*$ of the optimal policy in each time step $t \in [T]$

$$\theta_t^* = \arg\max_{\theta_t \in \Theta} \mathbb{E}_{x \sim \mathcal{X}, a \sim \pi} \Big[ r(x,a,t) - \tau \mathrm{D}_{\mathrm{KL}}[\pi_{\theta_t}(\cdot|x)\|\pi_{\mathrm{ref}}(\cdot|x)] \Big]. \tag{15}$$

We now introduce further assumptions on the setting. In order to make the learning process possible, we bound the 2-norm of the feature and parameter spaces.

**Assumption 1.** (Boundedness) The parameters and features are bounded: $\theta \in \Theta$ where $\Theta = \{\theta \in \mathbb{R}^d \mid \|\theta\|_2 \le W\}$ and $\Phi = \{\phi(x,a) \in \mathbb{R}^d \mid \|\phi(x,a)\|_2 \le L\}$.

It is known that an equivalence class of reward models leads to the same preferences under the Bradley-Terry model (Rafailov et al., 2024). This is similarly true in the case of the Dynamic Bradley-Terry model, because the implicit reward of NS-DPO, shown in Equation (8), relates the reward to the policy parameters $\theta$. We thus construct the following constraint on the policy class to properly specify the problem (Chowdhury et al., 2024).

**Assumption 2.** (Identifiability) The optimal policy in each time step $t$ corresponds to a single parameter in $\Theta$, which satisfies Equation (15): $\mathbf{1}_d^\mathsf{T} \theta_t^* = 0 \; \forall t \in [T]$, where $\mathbf{1}_d^\mathsf{T} \in \mathbb{R}^d$ is a vector of 1s.

We consider the setting where the true underlying parameter $\theta_t^* \in \Theta, \forall t \in [T]$ of the optimal policy $\pi^*$ is changing at each time step. We do not constrain how the optimal parameter changes, but instead upper bound the possible parameter drift allowed in the environment up to time step $T$. This upper bound is known as the variation budget.

**Assumption 3.** (Variation Budget Bound) The parameter drift of $\theta_t^* \in \Theta$ across $T$ timesteps is upper bounded as $\sum_{t=1}^{T-1} \|\theta_{t+1}^* - \theta_t^*\|_2 \le B_T$ where $B_T > 0$ is a known constant.

In the offline setting, our learning is constrained by the available dataset $\mathcal{D}$. A standard assumption in the offline learning literature is that of data coverage (Chowdhury et al., 2024; Zhu et al., 2023). The data coverage assumption ensures that the reference policy $\pi_{\mathrm{ref}}$ suitably explores the space of plausible responses of the optimal policy. We define the population covariance matrix as $\Sigma_\pi = \mathbb{E}[\phi(x,a)\phi(x,a)^\mathsf{T}] - \mathbb{E}[\phi(x,a)]\mathbb{E}[\phi(x,a)]^\mathsf{T}$, where the expectation is calculated over samples $x \sim \mathcal{X}, a \sim \pi(\cdot|x)$. The condition number $\kappa_\pi$ compares the coverage of the two policies $\pi$ and $\pi_{\mathrm{ref}}$

$$\forall \pi \in \Pi: \; \kappa_\pi = \sup_{v \in \mathbb{R}^d} \frac{v^\mathsf{T} \Sigma_\pi v}{v^\mathsf{T} \Sigma_{\pi_{\mathrm{ref}}} v} = \frac{\lambda_{\max}(\Sigma_\pi)}{\lambda_{\min}(\Sigma_{\pi_{\mathrm{ref}}})}, \tag{16}$$

while we use $\kappa = \max_\pi \kappa_\pi$ to denote the maximum value of $\kappa_\pi$. The definition of $\kappa_\pi$ requires that the reference policy sufficiently explores the feature space, which leads to the following assumption.

**Assumption 4.** (Feature Coverage) The reference policy $\pi_{\mathrm{ref}}$ satisfies $\lambda_{\min}(\Sigma_{\pi_{\mathrm{ref}}}) > 0$.

In a time-varying setting, the quality of the dataset $\mathcal{D}$ also depends upon its temporal coverage. We use the following assumptionm which also guarantees a minimal amount of data in each time step. Having enough data in each time step is motivated by the fact that we are assuming no knowledge of the dynamics of the actual preference drift. Note that $\Theta(T)$ in the assumption is the notation for the complexity, which is different from the parameter set $\Theta$ in Assumption 1.

**Assumption 5.** (Temporal Coverage) For each time step $t \in [T-1]$, the number of datapoints in the training set is between $\underline{m}$ and $\bar{m}$, where $\underline{m} > 0$ and $\bar{m} > \underline{m}$ are constants (i.e., $n = \Theta(T)$).

## 4.1 THEORETICAL RESULTS

**Estimation Error.** To bound the expected regret of the policy trained with NS-DPO, bounding the difference between the optimal and the learnt parameter is required. To analyse the parameter estimation error, we define the discounted covariance matrix of the offline dataset as

$$\hat{\Sigma} = \frac{1}{n} \sum_{i=1}^{n} \gamma^{T-t_i-1}(\phi(x_i, a_i) - \phi(x_i, a_i'))(\phi(x_i, a_i) - \phi(x_i, a_i'))^{\mathsf{T}}. \tag{17}$$

Under the assumptions from Section 4, we introduce bounds on the estimation error of the parameter $\tilde{\theta}_T$, which minimises the NS-DPO loss in Equation (12), with respect to the true parameter $\theta_T^*$ and $\hat{\Sigma}$:

$$\|\theta_T^* - \tilde{\theta}_T\|_{\hat{\Sigma}+\lambda I}, \tag{18}$$

where $\lambda > 0$ is introduced to guarantee the inversion of the matrix $\hat{\Sigma} + \lambda I$. The upper bound on the estimation error is shown in Theorem 1 and a detailed proof of the result is provided in Appendix E.1. Our analysis differs from the stationary case (Chowdhury et al., 2024), as we consider the temporally discounted datapoints in the NS-DPO loss. This is reflected in the covariance matrix $\hat{\Sigma}$ by the inclusion of the $\gamma^{T-t_i-1}$ term, which decreases the influence of observations that happened further in the past. As part of our analysis, we separate the estimation error into a *learning* term and *tracking* term. This tracking term accounts for the error introduced by the non-stationary nature of the environment, depending upon $B_T$ and the choice of $\gamma$ in the algorithm to upper bound it. We outline a suitable choice for $\gamma$ below.

**Theorem 1.** *(Estimation error of $\tilde{\theta}_T$.) Let $\delta \in (0,1], \lambda > 0, \tau > 0$. Let $\hat{\theta}_T$ denote the minimiser of the NS-DPO loss defined in Equation* (12)*. Let $\tilde{\theta}_T \in \Theta$ denote the parameter obtained by performing the parameter projection procedure on $\hat{\theta}_T$. Then with probability at least $1 - \delta$:*

$$\|\tilde{\theta}_T - \theta_T^*\|_{\hat{\Sigma}+\lambda I} \le 2\sqrt{\lambda}W + \underbrace{\frac{2C_1}{\tau c_{\sigma,\tau}}\sqrt{\frac{d + \log(1/\delta)}{n}}}_{\text{learning}} + \underbrace{\frac{16LR_{\sigma,\tau}\bar{m}}{T(1-\gamma)^{\frac{3}{2}}}\sqrt{\frac{d\bar{m}}{n}}B_T}_{\text{tracking}} \tag{19}$$

*where $C_1 > 0$ is a constant.*

**Expected Regret Bound.** Starting from the definition of the expected regret in Equation (13), the regret can be expressed with the estimation error in Equation (19). We then use our results in Theorem 1 to complete the analysis. The details of the regret analysis are deferred to Appendix E.2.

**Theorem 2.** *(Regret bound of $\tilde{\theta}_T$) Let $\delta \in (0, \frac{1}{2}], \tau > 0$. Let $\tilde{\theta}_T$ denote the parameter in $\Theta$ which minimises the NS-DPO loss (Equation* (12)*) on an offline dataset. The following bound holds with probability at least $1 - 2\delta$ and when $\lambda \ge C\sqrt{d\log(4d/\delta)/n}$:*

$$R_T^{\text{off}} \le \frac{\tau\kappa\bar{m}T(1-\gamma)}{2\underline{m}(1-\gamma^{T-1})}\left(2\sqrt{\lambda}W + \frac{2C_1}{\tau c_{\sigma,\tau}}\sqrt{\frac{d + \log(1/\delta)}{n}} + \frac{16LR_{\sigma,\tau}\bar{m}}{T(1-\gamma)^{\frac{3}{2}}}\sqrt{\frac{d\bar{m}}{n}}B_T\right)^2,$$

*where $C_1 > 0$ denotes a constant. When $\gamma = 1 - \left(\frac{B_T}{T}\right)^{3/4}$, $R_T^{\text{off}}$ satisfies:*

$$R_T^{\text{off}} = \tilde{O}\left(d\,B_T^{3/4}\,n^{-1/4}\right).$$

Standard offline bandits and RL algorithms assuming the stationarity of the underlying *scalar-valued reward* achieve $O(n^{-1/2})$ regret (Wang et al., 2021; Zhan et al., 2024; Qiao & Wang, 2024; Cen et al., 2024). For stationary preference-based rewards, Chowdhury et al. (2024) show an $O(n^{-1/4})$ regret/sub-optimality gap for DPO algorithm, whereas Nika et al. (2024) obtain an $O(n^{-1/2})$ regret. Unlike these prior work assuming stationary preferences, NS-DPO uses the discount weight $\gamma = 1 - \left(\frac{B_T}{T}\right)^{3/4}$ to address the non-stationarity in the dataset, which results in the regret bound above. However, our approach is general enough to capture the stationary setting, which corresponds to $B_T \to 0$. By setting $\gamma = 1 - \left(\frac{B_T}{T}\right)^{\alpha}$ with $0 < \alpha < \frac{2}{3}$, we show that the tracking term in the estimation error bound goes to zero. Corollary 3, shows that the widely considered stationary setting is a special case of NS-DPO. We provide the detailed proof in Appendix E.3.

**Corollary 3.** *(Regret bound under stationary preferences) Let $B_T \to 0$, $\delta \in (0, \frac{1}{2}], \tau > 0$. Let $\tilde{\theta}_T \in \Theta$ denote the minimiser of the NS-DPO loss (Equation (12)). Then, for $\lambda \geq C\sqrt{d \log(4d/\delta)/n}$, some constant $C_1 > 0$, $\gamma = 1 - \left(\frac{B_T}{T}\right)^{\alpha}$ and $0 < \alpha < 2/3$, we have with probability at least $1 - 2\delta$:*

$$\lim_{B_T \to 0} R_T^{\text{off}} < \frac{4\tau\kappa\bar{m}}{\underline{m}}\left(\sqrt{\lambda}W + \frac{C_1}{\tau c_{\sigma,\tau}}\sqrt{\frac{d + \log(1/\delta)}{n}}\right)^2,$$

*and recover the complexity of $R_T^{\text{off}} = O(n^{-\frac{1}{2}})$ under stationary preferences.*

## 5 EXPERIMENTS

In this section, we empirically evaluate NS-DPO's ability to learn under preference drift. We first show that NS-DPO outperforms DPO in the log-linear policy setting, supporting our theoretical results introduced in Section 4.1. We then analyse how NS-DPO performs under different types of preference drift and under different strengths of preference change using the Llama2 LLM (Touvron et al., 2023) and the Llama3 LLM (Dubey et al., 2024). We provide the code[2] used for the experiments.

### 5.1 EXPERIMENTAL SETUP

#### 5.1.1 SYNTHETIC EXPERIMENTS

To analyse the performance of NS-DPO in the log-linear policy class, we construct a synthetic environment with a known feature space and preference drift. We use the feature space from (Li et al., 2023), where $x \in \mathcal{X} = [0,1]^{d_x}$, $a \in \mathcal{A} = [n_a]$. The dimensions of the feature space and the policy parameter are both $2 \cdot d_x$. We use $d_x = 4, d_\theta = 8, |\mathcal{A}| = 16$ for all synthetic experiments.

**Non-stationary Dataset.** To construct a dataset $\mathcal{D} = \{x, a, a', t\}_{i=1}^n$, we randomly sample $x \sim X$ and $a_1, a_2 \sim \mathcal{A}$. We assign 20 datapoints per time step $\forall t \in [100]$. We sample 100 datapoints for evaluation at $T = 101$. To introduce preference drift, we follow an approach similar to Faury et al. (2021). We sample the preferences over $a_1$ and $a_2$ from the class of log-linear policies given in Equation (11), parameterised by $\theta_t^*$. We denote preferred response as $a$ and the rejected response as $a'$. When $t < 33$, we set the optimal parameter $\theta_t^* = (1,0,1,0,1,0,1,0)^\intercal$. For $t > 66$, we set $\theta_t^* = (0,1,0,1,0,1,0,1)^\intercal$. For $33 \leq t \leq 66$, we rotate $\theta_t^*$ smoothly between the two. For full details on the feature space and rotation see Appendix D.5.

**Algorithms for Synthetic Experiments.** We compare NS-DPO with DPO and SW-DPO in synthetic experiments. SW-DPO uses a "sliding" window to only consider points close to the current timestep $T$, which is commonly used in the non-stationary bandit literature (Garivier & Moulines, 2008). We test the performance of NS-DPO and SW-DPO over several values of $\gamma \in \{0.7, 0.9\}$ and window size $w \in \{33, 50\}$. The regularisation coefficient is $\tau = 1.0$ for all algorithms. We normalise the scale of the gradient for each method to address the differences caused by the application of exponential weighting and sliding window. For the reference policies, we use a uniform policy, whose parameter $\theta_{\text{ref}} \in \mathbb{R}^d$ is a zero vector.

**Evaluation Metrics.** To analyse the performance of the algorithms, we use the reward accuracy of the trained policies. The reward accuracy is computed by the portion of test response pairs with correctly estimated preferences, using the implicit rewards defined in Equation (8). For each tested algorithm, we report averaged results of the experiments across 10 different random seeds.

#### 5.1.2 LARGE LANGUAGE MODEL EXPERIMENTS

To test the performance of NS-DPO in an LLM setting, we create three preference datasets with known and controlled preference drift.

**Creating Non-Stationary Preference Datasets** To create datasets with varying preference drift, we select two reward models $r_1, r_2$ that result in different preferences for the responses $a$ and $a'$. We assign each datapoint an arbitrary time across 100 timesteps $t \in [100]$ and adjust the response

---

[2]https://anonymous.4open.science/r/ns-dpo-CD67/

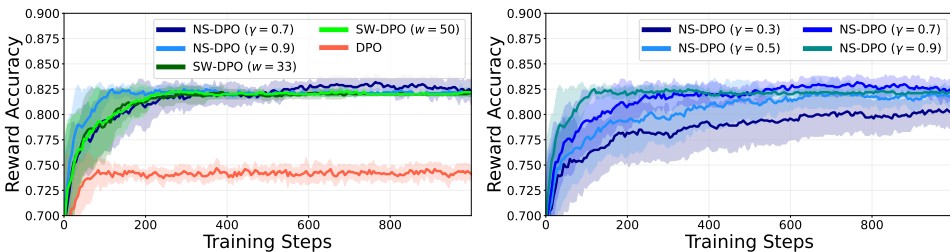

Figure 2: Synthetic experiment results with $d_x = 4, |\mathcal{A}| = 16$. The shaded area represents the standard deviation of each algorithm. [Left] NS-DPO and SW-DPO successfully addresses the non-stationarity present in the dataset, while stationary DPO fails to do so. NS-DPO shows faster training than SW-DPO, even compared to the case where the value of the window parameter $w$ for SW-DPO is set to the optimal value of 33. [Right] An ablation study on how different values of the discount factor $\gamma$ affect the training of NS-DPO. As the value of $\gamma$ becomes larger, the final test accuracy of the policy is achieved in fewer training steps.

preference according to two main modes of preference change, sudden or gradual. For sudden preference change, we select a change point $t_{cp} \in [100]$ for datapoints with a time before $t_{cp}$ we assign preferences based on $r_1$ and for points after $t_{cp}$ we assign preferences based on reward model $r_2$. For gradual preference change, we linearly interpolate the reward of each prompt response pair $(x, a)$ over some subset of the timesteps $T_{grad} \subset [100]$ (see Appendix D.2). Finally, we also adjust how the strength of preference change affects the performance of NS-DPO. We introduce $\rho_{diff}$, which is the portion of datapoints included in the dataset whose preferences change when assigning preferences according to $r_2$ instead of $r_1$. We provide further details in Appendix D.1.

**Datasets.** We created non-stationary preference datasets for the GlobalOpinionsQA dataset (Durmus et al., 2023) and Helpful Harmless dataset (Bai et al., 2022a) using the *helpsteer-helpfulness* and *beavertails-is_safe* outputs of the ARMORM model. We create two Ultrafeedback datasets (Cui et al., 2023), one using PAIRRM (Jiang et al., 2023) and ARMORM (Wang et al., 2024) reward models. We will make the datasets available as open-source.

**Language Models.** We use `Llama-2-7b-chat-hf` [3] and `Llama-3.2-1b-it` [4] (Touvron et al., 2023; Dubey et al., 2024) for both fine-tuning and the reference model. To reduce the compute demands of fine-tuning `Llama-2-7b-chat-hf`, we train LoRA weights (Hu et al., 2022) (see Appendix D.4 for further details). We fine-tune all parameters of `Llama-3.2-1b-it`.

**Evaluation Metrics.** To compare the performance of NS-DPO and the baseline algorithms in LLM datasets, we use reward accuracy. We also use Length Controlled Win Rate (LCWR) evaluated by AlpacaEval2 (Dubois et al., 2024) for experiments with `Llama-3.2-1b-it`.

**Algorithms for the LLM experiments.** We compare NS-DPO against baselines including stationary DPO and IPO. We also construct an In-Context Learning (ICL) algorithm referred to as tDPO, in which information about the time step is appended to the prompts of the data. All algorithms use the same supervised fine-tuned (SFT) model as the reference model. We use the SFT procedure from Rafailov et al. (2024), training the model on the preferred responses in the dataset. NS-DPO uses $\tau = 0.1$ and $\gamma = 0.95$ for fine-tuning `Llama-2-7b-chat-hf` with 2C NSGO dataset and UltraFeedback dataset. For Time Varying Helpful-Harmless (TV-HH) dataset, we adjust the value of $\gamma$ as $\gamma = 1 - (\frac{1}{100-t_{cp}})\log(100)$. For `Llama-3.2-1b-it`, we use $\tau = 1.0$ and $\gamma = 0.85$.

## 5.2 EXPERIMENT RESULTS

**How does NS-DPO perform when specialised to log-linear policy classes?** We present synthetic experiment results to compare the behaviour of NS-DPO and other algorithms with log-linear policies. As shown in the left image of Figure 2, when compared to NS-DPO and SW-DPO, DPO shows the worst performance with respect to the test data. Both NS-DPO and SW-DPO, which account for the preference drift present in the data, show significantly better performance. SW-DPO achieves similar performance to NS-DPO in the later stages of training, but NS-DPO achieves this performance in

---

[3]https://huggingface.co/meta-llama/Llama-2-7b-chat-hf

[4]https://huggingface.co/meta-llama/Llama-3.2-1B-Instruct

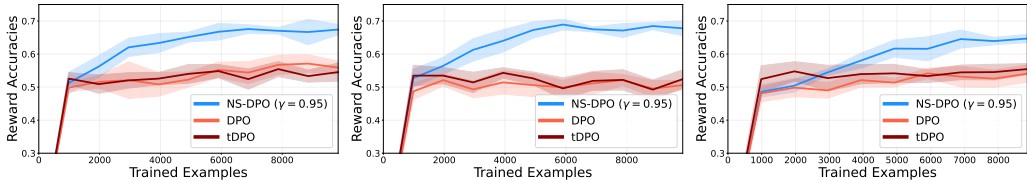

Figure 4: `Llama-2-7b-chat-hf` experiment results using 2C NSGO dataset. [Left] Opinion drift from the US to Germany. [Middle] Opinion drift from the US to Japan. [Right] Opinion drift from the US to Brazil. NS-DPO stays robust to the non-stationarity present in the dataset and achieves reward accuracies above 60%, while stationary methods show dropped reward accuracies of around 55%. Including the time steps in the prompt (tDPO) does not help meaningfully improve the performance of stationary DPO.

fewer training steps. As NS-DPO only varies the weights of datapoints, rather than removing them entirely, it can still leverage the information of datapoints in the earlier time steps. The right image of Figure 2 shows a comparison of different values of $\gamma$, ranging from 0.3 to 0.9. The results show that the performance of NS-DPO is stable in terms of the final test accuracy across a large range of values, $\gamma \in [0.5, 0.9]$. As the value of $\gamma$ is reduced, only points closest to the current time step contribute significantly to the gradient update of the model. Thus as $\gamma$ decreases, NS-DPO requires more training steps for the reward accuracy on the test set to converge.

**In summary:** NS-DPO outperforms the stationary DPO method, and achieves the same performance as other non-stationary baseline approaches in fewer training steps. The final performance of NS-DPO is robust to the value of $\gamma$ across a wide range of values.

**How robust and effective is NS-DPO under varying strengths of *sudden* preference drift?** We conduct two LLM experiments to investigate how varied strengths of sudden preference drift affect the NS-DPO's performance. Firstly, we vary $\rho_{\text{diff}}$, the portion of datapoints with preferences that change, at three different change points on the non-stationary UltraFeedback Dataset introduced in Figure 6. Secondly, we vary the change point for three different values of $\rho_{\text{diff}}$ on the TV-HH dataset. Stationary preference algorithms treat non-stationary preferences as label noise in the data. As $\rho_{\text{diff}}$ is increased, the level of noise observed by the stationary algorithms increase, leading to worse performance. We show this in Figure 10 and Figure 5 where for high values of $\rho_{\text{diff}}$, when the change

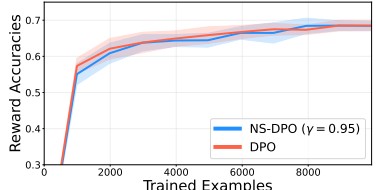

Figure 3: Training curves of NS-DPO and DPO trained with the UltraFeedback dataset without preference drift ($t_{\text{cp}} = 0$). `Llama-2-7b-chat-hf` is used. NS-DPO matches the performance of DPO even in stationary settings.

point is close to the present, the difference in performance between NS-DPO and the baseline algorithms can be as much as 20%. We also see NS-DPO outperfoming stationary DPO in Figure 6, where all the parameters of `Llama-3.2-1b-it` are fine-tuned. Datasets with a change point that occurs close to the present have very few examples of the new preference distribution. Because of this, stationary algorithms learn the old preference distribution, as that is mostly represented in the data. The low performance of the baseline algorithms on the binary classification of preferences at test time demonstrates this empirically. Note that the performance of NS-DPO matches that of DPO even when the preference shift in the dataset is not significant, $\rho_{\text{diff}} \leq 0.7$. This observation is further supported by Figure 3, where NS-DPO matches the performance of stationary DPO in a dataset with no preference drift. These results show that NS-DPO is robust against strong preference drift in offline datasets and matches the performance of stationary algorithms when the preference drift is trivial.

**In summary:** Standard preference learning approaches fail under strong preference drift, learning equally from old and recent preferences. NS-DPO is robust in these settings, and matches the performance of stationary approaches when the preference drift is small or non-existent.

**How does NS-DPO perform under *gradual* preference drifts?** Here we investigate how LLMs trained with NS-DPO perform when preference drift happens gradually over time. In Figure 7, we see that NS-DPO outperforms the DPO reward accuracy by over 10% on the TV-HH dataset with gradual preference drift. We note that the performance of NS-DPO is dependent upon the value of $\gamma$ chosen, however both approaches outperform the stationary baseline. The experiment results on the 2C NSGO dataset, which also simulates a gradual drift of preferences, are given in Figure 4. NS-DPO

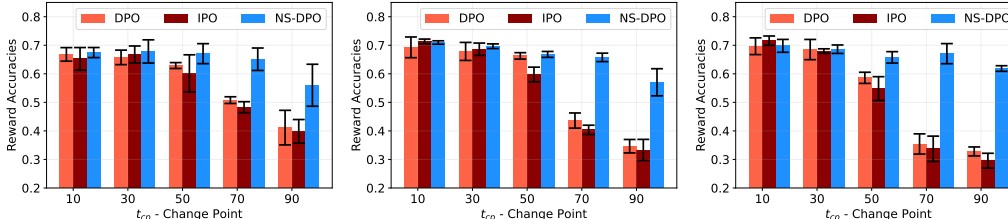

Figure 5: NS-DPO consistently outperforms DPO and IPO as the change point, $t_{cp}$ nears the present $T = 101$ for varying strengths of preference shift on the TV-HH dataset using the `Llama-2-7b-chat-hf` model. [Left] $\rho_{\text{diff}} = 0.7$. [Middle] $\rho_{\text{diff}} = 0.8$. [Right] $\rho_{\text{diff}} = 0.9$. We note that as the value of $t_{cp}$ increases, the performance difference between NS-DPO and the baselines increases. This is because as the change point moves closer to the present time step, the number of samples available from the updated preference distribution decreases. NS-DPO discounts samples with old preferences, focusing learning upon the small number of samples with up-to-date preference labels.

shows significantly better performance compared to stationary DPO, showing a performance gap of nearly 10% in reward accuracy. This difference is mainly caused by stationary methods failing to efficiently learn from datapoints at later time steps. tDPO, which trains the policy with time step information appended to the prompt, does not show a significant difference from stationary DPO.

**In summary:** NS-DPO outperforms stationary approaches when preferences change *gradually* over multiple time steps instead of at a specific change point.

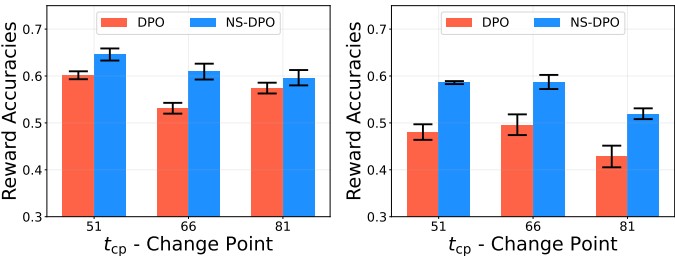

Figure 6: Full fine-tuning of `Llama-3.2-1b-it` with UltraFeedback-RM dataset, with $\rho_{\text{diff}} = 0.7$ on the left and $\rho_{\text{diff}} = 1.0$ on the right, with varying change points over 3 seeds.

## 6 CONCLUSION

In this work we propose NS-DPO, a practical and provably efficient approach for preference optimization on non-stationary offline datasets. With standard assumptions, we provide a theoretical analysis on the performance of NS-DPO in the case of log-linear policies. NS-DPO achieves a sample complexity of $O(n^{-1/4})$, and as $B_T \to 0$ the complexity of the regret recovers $O(n^{-1/2})$, found in the stationary setting. We further support this result with a suit of empirical results on a synthetic setting. We also investigate the application of NS-DPO to LLMs, create several non-stationary preference datasets, and show that NS-DPO shows superior performance to standard preference optimization algorithms and In Context Learning approaches on these datasets. Even in stationary settings, NS-DPO matches the performance of stationary algorithms. This

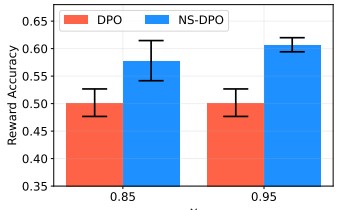

Figure 7: NS-DPO outperforms DPO in settings where preference drift occurs slowly across multiple timesteps. Here we compare NS-DPO and DPO on the TV-HH dataset with a gradual preference shift.

motivates the usefulness of our approach when the existence of preference drift in a dataset is unknown, as applying NS-DPO will not hurt performance even if the preference drift is too small to matter. Our approach can be easily extended to the online setting where data is sequentially provided as time passes. NS-DPO can also be adapted to learn at a time step that is not the present by discounting both past and future preference as a function of their distance from the time step of interest. We leave these ideas for future work.

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

## A  APPENDIX CONTENTS

In Appendix B, we provide further related works on DPO algorithms, different alignment settings, and a discussion of works that consider time varying alignment problems. Appendix C analyses the gradient of the NS-DPO objective. Appendix D explains the details of experiments conducted, including the creation of non-stationary datasets for LLM experiments and the behaviour of NS-DPO and SW-DPO in the synthetic setting. We provide proofs of our theoretical analysis in Appendix E step by step. In-depth derivations necessary for deriving the learning error are separately presented in Appendix E.4.

## B  FURTHER RELATED WORKS

Recent interest in the alignment of LLMs has lead to a wide variety of works. We briefly discuss further works that focus upon direct preference alignment algorithms.

Several approaches examine preference optimisation from a game theory perspective, avoiding the implicit assumptions of the BT model. In these settings the current policy plays against previous versions to further improve performance (Swamy et al., 2024; Rosset et al., 2024; Wu et al., 2024b; Yuan et al., 2024; Chen et al., 2024b; Pang et al., 2024; Munos et al., 2024). Xu et al. (2023) propose a cringe loss based objective whilst Hong et al. (2024); Pentyala et al. (2024); Hua et al. (2024) try to combine the supervised fine-tuning and preference optimization steps. Hong et al. (2024); Hua et al. (2024) propose a single training objective to do this and Pentyala et al. (2024) examine combining two different models trained on an SFT and direct preference objective respectively. Finally, Lu et al. (2024) propose a meta algorithm which uses an LLM to optimize the form of the direct preference learning objective itself.

An orthogonal direction of work is the online setting (Qi et al., 2024; Zhang et al., 2024; Guo et al., 2024; Xie et al., 2024), where feedback is returned by a human labeler or superior model. Khaki et al. (2024); Liu et al. (2024) adapt the offline settings using techniques such as rejection sampling to approximate an online setting. In this work we only consider the offline setting for simplicity, however the approach we propose can easily be adapted to the online setting. Other important directions of research include safety and robustness. Dai et al. (2024); Ramesh et al. (2024); Wu et al. (2024a) consider robust settings where safety or group information is known at training time and Dai et al. (2024) analyse a constrained optimization problem through the lens of safety in LLMs. Whilst these approaches look to address a wide range of settings, our work is the first to provide a solution to the case of non-stationary preferences.

Carroll et al. (2024) consider how to correctly align LLMs under preference drift, showing several possible goals for alignment in an online setting. Whilst in the online non-stationary setting the LLM can adapt to the changing preferences of the user, our setting considers aligning the model on an offline dataset before deploying the static model to users at test time. As such our approach is most similar to the *Privileged Reward* and *Initial Reward* settings Carroll et al. (2024) proposes, as we determine that the preferences exhibited in the present are the most important (*Privileged Reward*) and future users will interact with a model aligned to preferences from their past (*Initial Reward*).

## C  ANALYSIS OF NS-DPO GRADIENT

Here we analyse the gradient of the NS-DPO loss objective. The gradient of Equation (10) with respect to the model parameters $\theta$ is as follows:

$$\nabla_\theta \mathcal{L}^{\mathrm{NS}}(\theta) = \sum_{(x_i, a_i, a_i', t_i) \in \mathcal{D}} \underbrace{-\tau \gamma^{T-t_i-1} \sigma\left(-h_\theta(x_i, a_i, a_i')\right)}_{\text{Gradient scaling}} \underbrace{\left(\nabla_\theta \log \pi_\theta(a_i|x_i) - \nabla_\theta \log \pi_\theta(a_i'|x_i)\right)}_{\text{Gradient Direction}}.$$

(20)

The gradient of the NS-DPO objective consists of two terms. The first term $\sigma\left(-h_\theta(x_i, a_i, a_i')\right)$ scales the gradient update, which increases when the model incorrectly prefers response $a_i'$ to $a_i$ and decreases when the model correctly predicts the response preference. **NS-DPO only adjusts the scaling term** of the gradient by discounting the scaling term further when points are temporally far away from $T$. The second term, $\nabla_\theta \log \pi_\theta(a_i|x_i) - \nabla_\theta \log \pi_\theta(a_i'|x_i)$, controls the direction of the gradient update.

In the case of stationary preferences in the dataset (points whose preference does not change at any time $t_i$), the gradient of these points is still applied to the parameters $\theta$ by the NS-DPO Loss with scaling by the term $\gamma^{T-t_i-1}$. Whilst this downweights these gradients this is price of not knowing which points have changing preferences and which points have fixed preferences within our setting. When we know that there is no preference drift, we set the value of $\gamma$ to 1 to remove discounts (see Appendix E.3).

# D  FURTHER EXPERIMENT DETAILS

## D.1  CONTROLLING THE STRENGTH OF PREFERENCE DRIFT

In this section, we give more details on how $\rho_{\text{diff}}$ is calculated, which is used to control the degree of preference drift as reward models are changed in the experiments. We first note that when $t < t_{\text{cp}}$, *old* reward model is used to evaluate the preference of the given prompt-response pair, while we use *new* reward model to evaluate datapoints with $t \geq t_{\text{cp}}$:

$$r(x, a, t) = \begin{cases} r^{\text{old}}(x, a), & \text{if } t < t_{\text{cp}} \\ r^{\text{new}}(x, a), & \text{if } t \geq t_{\text{cp}}. \end{cases}$$

We then use $o_i^{\text{old}}$ and $o_i^{\text{new}}$ to denote the preference given by old and new reward model respectively, on the response pairs $(a_i, a_i')$ of prompt $x_i$:

$$o_i^{\text{old}} \sim \sigma(r^{\text{old}}(x_i, a_i) - r^{\text{old}}(x_i, a_i')),$$
$$o_i^{\text{new}} \sim \sigma(r^{\text{new}}(x_i, a_i) - r^{\text{new}}(x_i, a_i')).$$

Using $o_i^{\text{old}}$ and $o_i^{\text{new}}$, we calculate the portion of datapoints whose preferences differ between the old and new reward models:

$$\rho_{\text{diff}} = \frac{1}{n} \sum_{i}^{n} \mathbb{1}(o_i^{\text{old}} \neq o_i^{\text{new}}). \tag{21}$$

If the value of $\rho_{\text{diff}}$ is large, it means that the preference drift from the old reward model to the new reward model is happening stronger in the dataset. When $t_{\text{cp}}$ is fixed for the dataset, which means that the number of datapoints from each reward model is fixed, datasets with higher $\rho_{\text{diff}}$ will result in worse performance of the algorithms. This is because more datapoints evaluated with the old reward model will have conflicting preference with the new reward model, causing harm to learning the true preference.

### D.2 NON-STATIONARY PREFERENCE DATASET CREATION

**1) NSGO Datasets.** We modify the GlobalOpinionQA dataset[5] (Durmus et al., 2023) to create a time varying dataset. GlobalOpinionQA consists of questions regarding global issues, different responses, and preferences from several countries represented as a probability vector. We copy the questions and responses to create multiple time steps $t \in [100]$. We then vary the preferences with time by linearly interpolating between the preferences of two different countries. This simulates gradual preference drifts that can be caused by demographic shift or a series of external events. We generate preference drift using three pairs of countries. In each pair the starting country is the US, and the ending country is either Brazil, Japan or Germany. The preferences at the first and last time step correspond to either country in the pair. The last time step is held out as a test dataset and treated as the current time $T = 101$. We divide the prompt-response pairs so that training and test data do not share any prompts.

**2) UltraFeedback-RM Datasets.** Using the prompts and response candidates of UltraFeedback[6] (Cui et al., 2023), we obtain preferences from two different reward models, PAIRRM[7] (Jiang et al., 2023) and ARMORM[8] (Wang et al., 2024). The datapoints in the training set are randomly assigned to one of $t \in [100]$ time steps, and assigned preferences of PAIRRM if the time step $t$ is earlier than the change point $t_{\mathrm{cp}} \in \{51, 66, 81\}$. We assign the preferences of ARMORM for the datapoints with time steps $t \geq t_{\mathrm{cp}}$

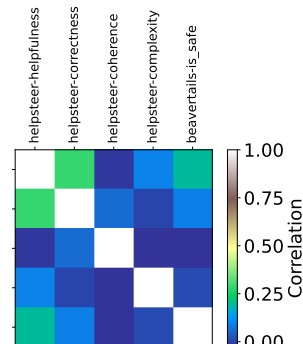

Figure 8: The correlation of different preference labels generated by rewards from the ARMORM reward model on the Helpful Harmless *harmless-base* dataset (Bai et al., 2022a). We observed that concepts such as safety and helpfulness have more correlated preferences, whilst the *helpsteer-coherence* reward model is un-correlated with the other models we analysed.

and datapoints in the test set with $T = 101$. To test the effect of varied degrees of preference drift, we also vary the portion of datapoints whose preferences flip as reward model changes. We denote this portion as $\rho_{\mathrm{diff}}$ and use $\rho_{\mathrm{diff}} \in \{0.7, 0.9, 0.95, 1.0\}$ to create both training and test data. We use 10k datapoints for training and 500 datapoints for testing.

**3) UltraFeedback-LM Datasets.** Using the same UltraFeedback dataset as above, we construct another dataset with the information of language models used for generations. The datapoints in the training set are randomly assigned to one of $t \in [100]$ time steps. Among the datapoints whose time step is earlier than the change point $t_{\mathrm{cp}} \in \{21, 51, 81\}$, $\rho_{\mathrm{diff}} \in \{0.7, 1.0\}$ of the datapoints have responses that are generated by *smaller* language models as preferred responses. The other datapoints have responses generated by `gpt-4` as preferred. We use 23.3k datapoints for training. We use the generations of `starchat`, `llama-2-7b-chat`, `wizardlm-7b`, `pythia-12b`, `alpaca-7b`, `llama-2-13b-chat`, `wizardlm-13b`, `ultralm-13b` for *smaller* language models in the dataset.

**4) Time Varying Helpful Harmless Datasets.** Using the *harmless-base* subset of the Helpful Harmless dataset[9] (Bai et al., 2022a), we create a time varying preference dataset. To do so, we use two reward models, the *helpsteer-helpfulness* and *beavertails-is_safe* outputs from the ARMORM model (Wang et al., 2024). Figure 8 shows that these rewards result in different preferences on the *harmless-base* dataset. We then assign each datapoint in the dataset a random time value from $t \in [100]$. We construct two methods to assign preferences using the time step information: change point preference shift and gradual variation. Under the change point preference shift, datapoints are assigned preferences according to *helpsteer-helpfulness* before the change point $t_{cp}$ and *beavertails-is_safe* after the change point. Under gradual variation, we use the following reward model

$$r(x, y, t) = \begin{cases} r_0(x, y) & t < 33 \\ r_0(x, y)\frac{(t-33)}{33} + r_1(x, y)\big(1 - \frac{t-33}{33}\big) & 33 \leq t < 66 \\ r_1(x, y) & t \geq 66, \end{cases}$$

---

[5] https://huggingface.co/datasets/Anthropic/llm_global_opinions

[6] We modify the binarized version of UltraFeedback.

[7] https://huggingface.co/llm-blender/PairRM

[8] https://huggingface.co/RLHFlow/ArmoRM-Llama3-8B-v0.1

[9] https://huggingface.co/datasets/Anthropic/hh-rlhf

where $r_0$ is the *helpsteer-helpfulness* reward and $r_1$ is the *beavertails-is_safe* reward. We use this type of schedule for gradual change to simulate preference drifts that happens gradually over a finite time horizon. We use $15k$ points for training and $2k$ for testing. We use reward models for helpfulness and safety, as these are both desired properties of an LLM but often result in differing preferences; for example, rewarding helpfulness can often lead to unsafe outputs when an LLM is asked a dubious question, like how to best rob a store.

### D.3 THE TWO COUNTRIES (2C) NON-STATIONARY GLOBAL OPINIONS DATASET

To test NS-DPO, we create a synthetic non-stationary dataset in which the temporal trends are known. To do this, we use the GlobalOpinionsQA dataset (Durmus et al., 2023). We preprocess the dataset in three major ways.

**Binary Preferences.** We convert the dataset to a dataset of binary preferences. For each set of prompt and responses, we create a row for each possible combination of prompt and binary response pairs. We calculate the preference probability for these response pairs as follows. Assuming the non-binary responses follow a Plackett-Luce preference framework, we can find the reward associated with responses (up to an additive constant) by taking the log of the preference probability. We can then take the sigmoid of these responses to find a normalised binary preference.

**Country Filter.** We filter the dataset down to the following countries: Nigeria, Egypt, India, China, Japan, Germany, France, Spain, United States, Canada, Brazil, Argentina, Australia and New Zealand.

**Country Level Prompts.** We filter the dataset such that each row of the dataset is the prompt, response, preference probability of a single country.

After the preprocessing, we copy the dataset and assign a different timestep to each unique instance of (prompt, response, preference). We simulate the drift in preferences by using preference probabilities of two countries, shifting from one to another over time. Out of $100$ time steps in the training dataset, the first 33 time steps consisted of preference probabilities from the US. Preference labels sampled from the last 33 time steps are from probabilities of the target country. We use Germany, Japan and Brazil as target countries, creating three different datasets. In the intermediate 33 time steps, preference labels are sampled from interpolated probabilities between these two countries. To introduce sufficient shift in preferences, we selected responses in which probabilities for the same response from two countries differed at least by $0.2$. We subsampled prompt-response pairs down to 10,000 datapoints, allowing each time step to consist of different prompts and responses. For evaluation, we used prompts and response candidates that are not present in the training data.

### D.4 COMPUTE RESOURCES USES

To run the LLM experiments, we use A100 GPUs with 40GB VRAM. The synthetic experiments are run locally on a laptop without using GPUs.

### D.5 SYNTHETIC EXPERIMENTS

We give further details about the setting of synthetic experiments. To analyse the performance of NS-DPO in the log-linear policy class, we construct a synthetic environment with a known feature space and preference drift. We use the feature space from (Li et al., 2023), where $x \in \mathcal{X} = [0,1]^{d_x}$, $a \in \mathcal{A} = [n_a]$ and $\phi(x,a)$ is computed as

$$\phi(x,a) = \left[(a+1)\cdot\cos(x_0\cdot\pi),\ \frac{1}{a+1}\cdot\sin(x_0\cdot\pi),\cdots,(a+1)\cdot\cos(x_{d_x-1}\cdot\pi),\ \frac{1}{a+1}\cdot\sin(x_{d_x-1}\cdot\pi)\right].$$
(22)

The dimensions of the feature space and the policy parameter are both $2 \cdot d_x$. We use $d_x = 4, d_\theta = 8, |\mathcal{A}| = 16$ for all synthetic experiments.

**Non-stationary Dataset.** To construct a dataset $\mathcal{D} = \{x, a, a', t\}_{i=1}^{n}$, we randomly sample $x \sim X$ and $a_1, a_2 \sim \mathcal{A}$. We assign 20 datapoints per time step $\forall t \in [100]$. We sample 100 datapoints for evaluation at $T = 101$. To introduce preference drift, we follow an approach similar to Faury et al. (2021). We sample the preferences over $a_1$ and $a_2$ from the class of log-linear policies given in Equation (11), parameterised by $\theta_t^*$. We denote preferred response as $a$ and the rejected response as

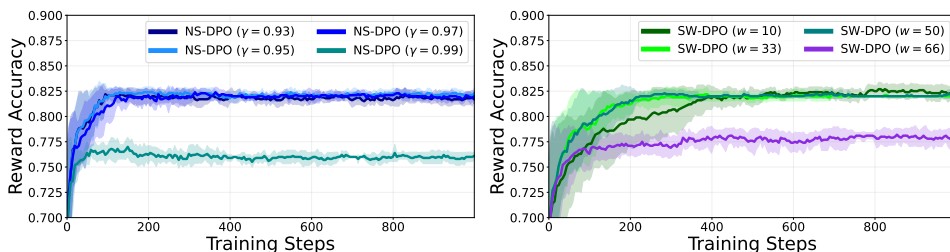

Figure 9: [Left] Performance of NS-DPO with values of $\gamma > 0.9$. NS-DPO shows robust performance with respect to the value of $\gamma$, while it starts resembling the performance of stationary DPO as the value approaches very close to 1, $\gamma > 0.97$. [Right] Expected RLHF objective gap of SW-DPO in the same experiments. The performance of SW-DPO improves as the value of $w$ gets closer to 33, when the algorithm is only learning from datapoints where the preference distribution stays stationary in the given setting. The setting with $w = 10$ also shows final performance similar to the case of $w = 33$, but it shows slower training because of the reduced amount of data used for training.

$a'$. When $t \leq 33$, we set the optimal parameter as $\theta_t^* = (1, 0, 1, 0, 1, 0, 1, 0)^\intercal$. Between $34 \leq t \leq 66$, the parameter $\theta_t^*$ varies as

$$\theta_t^* = \left[\cos(\tfrac{t-33}{33} \cdot \tfrac{\pi}{2}), \sin(\tfrac{t-33}{33} \cdot \tfrac{\pi}{2}), \ldots, \cos(\tfrac{t-33}{33} \cdot \tfrac{\pi}{2}), \sin(\tfrac{t-33}{33} \cdot \tfrac{\pi}{2})\right]^\intercal. \tag{23}$$

For the remaining time steps $67 \leq t \leq 100$, we use $\theta_t^* = (0, 1, 0, 1, 0, 1, 0, 1)^\intercal$.

**Further Results of NS-DPO and SW-DPO.** We present the experiment results of NS-DPO and SW-DPO on the synthetic dataset with varied values of hyperparameters $\gamma$ and $w$. As shown in Figure 9, The performance of NS-DPO is robust across varied values of $\gamma$, maintaining its reward accuracy over 80% when $0.5 \leq \gamma \leq 0.97$. In the case of SW-DPO, the performance is more sensitive to the change of the window size $w$. When $w = 10$, it shows similar test performance in the later stage of the training, while the process is visibly slowed down due to the reduced amount of datapoints actually being used. On the other hand, as the window size gets bigger and starts including datapoints where parameter shift introduces conflicting preferences, SW-DPO also shows degrading performance. These results provide further support the advantages of using NS-DPO over SW-DPO, as it shows faster training and less sensitivity to the hyperparameter.

## D.6 FURTHER RESULTS OF LLM EXPERIMENTS

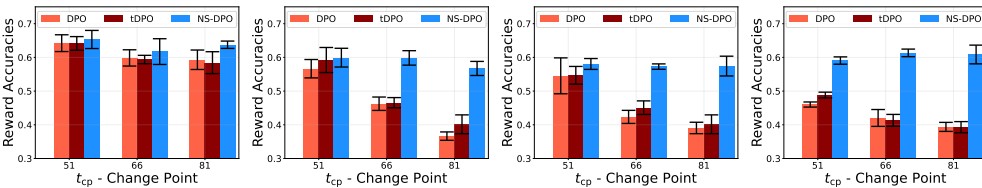

Figure 10: Experiment results conducted on UltraFeedback-RM dataset with preference drift.[Left] $\rho_{\text{diff}} = 0.7$. [Center Left] $\rho_{\text{diff}} = 0.9$. [Center Right] $\rho_{\text{diff}} = 0.95$. [Right] $\rho_{\text{diff}} = 1.0$. As $\rho_{\text{diff}}$, the percentage of training datapoints with flipped preference increases, DPO fails to learn the preference distribution at $T = 101$. Meanwhile, NS-DPO shows robust performance under various values of $\rho_{\text{diff}}$, maintaining reward accuracies above 50%. As $t_{\text{cp}}$, the change point of the reward model happens later in time, the gap between stationary approaches and NS-DPO gets larger. The experiments are run under a reward model shift from PAIRRM to ARMORM. `Llama-2-7b-chat-hf` is used, and the training dataset consists of 100 time steps.

For experiments with UltraFeedback-LM datasets, we use the length-controlled win rate (LCWR) of AlpacaEval2 (Dubois et al., 2024) for evaluating test performance. As shown in Table 1, NS-DPO shows higher LCWR than stationary DPO under various settings of preference drift. Even under no preference drift, which correspond to $\rho_{\text{diff}} = 0$ and $t_{\text{cp}} = 0$, NS-DPO shows better performance than stationary DPO.

|  |  | LCWR | |  |  | LCWR | |
|---|---|---|---|---|---|---|---|
| $\rho_{\text{diff}}$ | $t_{\text{cp}}$ | NS-DPO | DPO | $\rho_{\text{diff}}$ | $t_{\text{cp}}$ | NS-DPO | DPO |
| 0.7 | 21 | **8.93** | 7.29 | 1.0 | 21 | **9.00** | 8.23 |
| 0.7 | 51 | **8.38** | 7.85 | 1.0 | 51 | **7.41** | 6.99 |
| 0.7 | 81 | **7.85** | 7.17 | 1.0 | 81 | **7.36** | 6.49 |
| 0 | 0 | **9.12** | 8.81 | - | - | - | - |

Table 1: Length-Controlled Win Rates (LCWRs) of `Llama-3.2-1b-it` models, evaluated by AlpacaEval2. The models are trained with UltraFeedback-LM dataset (See Appendix D.2). NS-DPO outperforms stationary DPO under various types of sudden preference drift, with higher preference by GPT-4 evaluator.

## D.7 KL DIVERGENCE EVALUATION

| $\rho_{\text{diff}}$ | $t_{\text{cp}}$ | Approx. $D_{\text{KL}}[\pi_{\text{NSDPO}}\|\pi_{\text{ref}}]$ | Approx. $D_{\text{KL}}[\pi_{\text{DPO}}\|\pi_{\text{ref}}]$ |
|---|---|---|---|
| 0.7 | 51 | $0.375329 \pm 0.009827$ | $1.667666 \pm 0.034786$ |
| 0.7 | 66 | $1.253911 \pm 0.015404$ | $3.139016 \pm 0.113256$ |
| 0.7 | 81 | $0.296458 \pm 0.013328$ | $3.247607 \pm 0.022891$ |
| 1.0 | 51 | $-0.509829 \pm 0.024738$ | $1.301836 \pm 0.020247$ |
| 1.0 | 66 | $0.640597 \pm 0.010225$ | $0.881228 \pm 0.023252$ |
| 1.0 | 81 | $0.813176 \pm 0.018992$ | $2.043928 \pm 0.070260$ |

Table 2: KL divergence of `Llama-3.2-1b` models trained with NS-DPO and stationary DPO, with respect to the SFT reference model. We use three seeds per setting. Two columns on the left show the parameters used for generating time-varying UltraFeedback dataset. NS-DPO consistently shows lower KL divergence compared to stationary DPO.

We investigate the deviation of parameters caused by training with NS-DPO and stationary DPO. We fine-tune `Llama-3.2-1b` with stationary DPO and NS-DPO, using the time-varying modification of UltraFeedback dataset described in Appendix D.2. We evaluate the approximated KL divergence between the fine-tuned policy $\pi$ and the reference policy $\pi_{\text{ref}}$. For each prompt $x_i$, we sample 32 responses $a_j^i, j \in \{1, \ldots, 32\}$ from $\pi$ with maximum 32 tokens each. During sampling responses, each token is sampled from 50 candidates with highest probabilities. We then compute $\frac{1}{32} \sum_{j=1}^{32} \left( \log \pi(a_j^i|x_i) - \log \pi_{\text{ref}}(a_j^i|x_i) \right)$ to approximate the KL divergence $D_{\text{KL}}[\pi(\cdot|x_i)\|\pi_{\text{ref}}(\cdot|x_i)]$ of the policy with respect to $x_i$. We obtain $r(x_i, a_i) - \tau D_{\text{KL}}[\pi(\cdot|x_i)\|\pi_{\text{ref}}(\cdot|x_i)]$ per prompt as in Equation (7), compare the values of policies' generated outputs. This way, we determine which policy's response has won for the given prompt and evalute the win rate.

# E  OFFLINE LEARNING ANALYSIS

In this section, we provide the remaining details of the analysis on the offline learning of non-stationary dataset.

**Non-Linearity Coefficients.** Following the analysis from Filippi et al. (2010); Faury et al. (2021), we capture the non-linearity of the sigmoid function in the NS-DPO loss. We use the coefficients $k_{\sigma,\tau}, c_{\sigma,\tau}$, which are the supremum and infimum of $\dot{\sigma}(\tau\langle\phi(x,a) - \phi(x,a'),\theta\rangle)$ over $x \in \mathcal{X}, (a,a') \in \mathcal{A}^2, \theta \in \Theta$ respectively:

$$k_{\sigma,\tau} = \sup_{x\in\mathcal{X},(a,a')\in\mathcal{A}^2,\theta\in\Theta} \dot{\sigma}(\tau\langle\phi(x,a) - \phi(x,a'),\theta\rangle), \tag{24}$$

$$c_{\sigma,\tau} = \inf_{x\in\mathcal{X},(a,a')\in\mathcal{A}^2,\theta\in\Theta} \dot{\sigma}(\tau\langle\phi(x,a) - \phi(x,a'),\theta\rangle), \tag{25}$$

while we use $R_{\sigma,\tau} = k_{\sigma,\tau}/c_{\sigma,\tau}$ to denote the ratio between $k_{\sigma,\tau}$ and $c_{\sigma,\tau}$.

**Loss and gradient.** We recap the loss of NS-DPO with $\ell_2$ regularisation term:

$$\mathcal{L}_{\text{reg}}^{\text{NS}}(\theta) = -\frac{1}{n}\sum_{i=1}^{n}\left[\gamma^{T-t_i-1}\left\{o_i\log\sigma(h_\theta(x_i,a_i,a_i')) + (1-o_i)\log\sigma(h_\theta(x_i,a_i',a_i))\right\}\right] + \frac{\lambda c_{\sigma,\tau}\tau^2}{2}\|\theta\|^2. \tag{26}$$

We use Equation (26) to draw parallels between the NS-DPO objective in Equation (10) and the logistic regression objective used in the generalised linear bandit setting of (Faury et al., 2021). We assume the preference label $o_i$ is sampled from a Dynamic Bradley-Terry model with the true unknown environment parameter $\theta_{t_i}^*$. Under this assumption, the mean of the preference label is $\mathbb{E}[o_i|\{x_i,a_i,a_i',t_i\}] = \sigma(h_{\theta_{t_i}^*}(x_i,a_i,a_i'))$. When there is only a unilateral preference sampled for a given prompt-response pairs, the sigmoid function forces the implicit rewards of DPO to have infinitely large scale, driving $p(a \succ a')$ to either 1 or 0 (Azar et al., 2024). The $\ell_2$ regularisation term in our analysis mitigates this problem, by controlling the parameter norm. Differentiating Equation (12) with respect to the parameter $\theta$ results in

$$\nabla_\theta\mathcal{L}_{\text{reg}}^{\text{NS}}(\theta) = -\frac{1}{n}\sum_{i=1}^{n}\tau\gamma^{T-t_i-1}o_i\hat{\phi}_i + \underbrace{\frac{1}{n}\sum_{i=1}^{n}\left[\tau\gamma^{T-t_i-1}\sigma(h_\theta(x_i,a_i,a_i'))\hat{\phi}_i\right] + \lambda c_{\sigma,\tau}\tau^2\theta}_{:=g^\tau(\theta)}, \tag{27}$$

where $\hat{\phi}_i = \phi(x_i,a_i) - \phi(x_i,a_i')$ is also introduced for brevity. We denote the parameter-dependent part of the gradient as $g^\tau(\theta) = \frac{1}{n}\sum_{i=1}^{n}\left[\tau\gamma^{T-t_i-1}\sigma(h_\theta(x_i,a_i,a_i'))\hat{\phi}_i\right] + \lambda c_{\sigma,\tau}\tau^2\theta$ which we will use to analyse the parameter estimation error.

**Parameter Projection.**  Let $\hat{\theta}_T$ denote the parameter minimising the NS-DPO loss defined in Equation (12), $\hat{\theta}_T = \arg\min_{\theta\in\mathbb{R}^d}\mathcal{L}^{\text{NS}}(\theta)$. Due to both learning and tracking aspects of the estimation error, we cannot guarantee that $\hat{\theta}_T$ is within the boundary of the parameter presented in Assumption 1, $\hat{\theta}_T \in \Theta$. This motivates a parameter projection method, which enables finding an admissible parameter $\tilde{\theta}_T \in \Theta$ while minimising its deviation from $\hat{\theta}_T$ (Faury et al., 2021; Wang et al., 2023). Using $\tilde{\theta}_T$ in the performance analysis of NS-DPO allows preventing the potential violation of Assumption 1 when $\hat{\theta}_T$ is used. We perform parameter projection by calculating $\hat{\theta}_T$ by

$$\tilde{\theta}_T = \arg\min_{\theta\in\Theta}\|g^\tau(\hat{\theta}_T) - g^\tau(\theta)\|_{(\hat{\Sigma}+\lambda I)^{-1}}, \tag{28}$$

using $\hat{\Sigma}$ defined in Equation (17) and $g^\tau(\theta)$ defined in Equation (27).

**Covariance matrices.** In addition to $\hat{\Sigma}$ defined in Equation (17) we also define $\tilde{\Sigma}$, to which squared discount weights are applied:

$$\tilde{\Sigma} = \frac{1}{n}\sum_{i=1}^{n}\gamma^{2T-2t_i-2}(\phi(x_i,a_i) - \phi(x_i,a_i'))(\phi(x_i,a_i) - \phi(x_i,a_i'))^\intercal. \tag{29}$$

Due to its squared application of the exponential weighting, $\hat{\Sigma} \succ \tilde{\Sigma}$.

### E.1 ESTIMATION ERROR

**Theorem 1.** *(Estimation error of $\tilde{\theta}_T$.) Let $\delta \in (0, 1], \lambda > 0, \tau > 0$. Let $\hat{\theta}_T$ denote the minimiser of the NS-DPO loss defined in Equation (12) on an offline dataset. Let $\tilde{\theta}_T$ denote the parameter obtained by performing the parameter projection procedure on $\hat{\theta}_T$. Then with probability at least $1 - \delta$:*

$$\|\tilde{\theta}_T - \theta_T^*\|_{\hat{\Sigma}+\lambda I} \leq 2\sqrt{\lambda}W + \underbrace{\frac{2C_1}{\tau c_{\sigma,\tau}}\sqrt{\frac{d + \log(1/\delta)}{n}}}_{learning} + \underbrace{\frac{16LR_{\sigma,\tau}\bar{m}}{T(1-\gamma)^{\frac{3}{2}}}\sqrt{\frac{d\bar{m}}{n}}B_T}_{tracking} \tag{30}$$

*where $C_1 > 0$ is a constant.*

Estimation errors in typical stationary settings can be considered as *learning* errors, which are caused by having finite data sampled stochastically. In time-varying settings, the parameter estimation suffers from *tracking* error as well, which is caused by the drift of the underlying true parameter along the time steps (Faury et al., 2021; Wang et al., 2023). In this section, we show how these errors can be disentangled and bounded separately. To do this, we apply the approach of (Wang et al., 2023) in contextual bandit setting to our setting of offline preference learning.

#### E.1.1 BOUND DECOMPOSITION

We begin with the deviation between the optimal parameter $\theta_T^*$ and $\tilde{\theta}_T$, the projected parameter of the NS-DPO estimator $\hat{\theta}_T$:

$$g^\tau(\tilde{\theta}_T) - g^\tau(\theta_T^*) = \frac{1}{n}\sum_{i=1}^n \tau\gamma^{T-1-t_i}\left[\sigma(h_{\tilde{\theta}_T}(x_i, a_i, a_i')) - \sigma(h_{\theta_T^*}(x_i, a_i, a_i'))\right]\hat{\phi}_i + \lambda c_{\sigma,\tau}\tau^2(\tilde{\theta}_T - \theta_T^*). \tag{31}$$

Applying the mean value theorem to the difference of sigmoid functions in Equation (31) we get

$$g^\tau(\tilde{\theta}_T) - g^\tau(\theta_T^*) = \frac{1}{n}\sum_{i=1}^n \tau^2\gamma^{T-1-t_i}\left[\int_{v=0}^1 \dot{\sigma}(\tau\langle\hat{\phi}_i, (1-v)\theta_T^* + v\tilde{\theta}_T\rangle)dv\right]\hat{\phi}_i\hat{\phi}_i^\intercal(\tilde{\theta}_T - \theta_T^*)$$
$$+ \lambda c_{\sigma,\tau}\tau^2(\tilde{\theta}_T - \theta_T^*).$$

We can now define a matrix $\mathbf{G}_T$ to define the relation between $g^\tau(\tilde{\theta}_T) - g^\tau(\theta_T^*)$ and $\tilde{\theta}_T - \theta_T^*$:

$$\mathbf{G}_T := \frac{1}{n}\sum_{i=1}^n \gamma^{T-1-t_i}\underbrace{\left[\int_{v=0}^1 \dot{\sigma}(\tau\langle\hat{\phi}_i, (1-v)\theta_T^* + v\tilde{\theta}_T\rangle)dv\right]}_{\alpha(i,\theta_T^*,\tilde{\theta}_T)}\hat{\phi}_i\hat{\phi}_i^\intercal + \lambda c_{\sigma,\tau}I, \tag{32}$$

$$g^\tau(\tilde{\theta}_T) - g^\tau(\theta_T^*) = \tau^2 \cdot \mathbf{G}_T \cdot (\tilde{\theta}_T - \theta_T^*). \tag{33}$$

We make a brief aside to show $\mathbf{G}_T \succeq c_{\sigma,\tau}(\hat{\Sigma} + \lambda I) \succeq 0$ (Faury et al., 2020; Filippi et al., 2010), as this is an important property of $\mathbf{G}_T$ and one we will use later in the main proof. To prove this, we first show that $\alpha(i, \theta_T^*, \tilde{\theta}_T) > c_{\sigma,\tau}$. $\alpha(i, \theta_1, \theta_2)$ is the mean value of $\dot{\sigma}$ along the path between some points $\langle\hat{\phi}, \theta_1\rangle$ and $\langle\hat{\phi}, \theta_2\rangle$. This is greater than the infimum of $\dot{\sigma}$ at a point along that path, which is in turn greater than the infimum of $\dot{\sigma}$ in the space of parameters $\theta \in \Theta$. The last infimum is the definition of $c_{\sigma,\tau}$ Equation (25). Then

$$\alpha(i, \theta_1, \theta_2) = \int_{v=0}^{v=1} \dot{\sigma}(\tau(v\phi_i^\intercal\theta_1 - (1-v)\phi_i^\intercal\theta_2))dv \geq \inf_{c\in[\phi_i^\intercal\theta_1, \phi_i^\intercal\theta_2]}[\dot{\sigma}(c)]$$
$$\geq \inf_{\phi\in\Phi, \theta\in\Theta}[\dot{\sigma}(\tau\phi^\intercal\theta)] = c_{\sigma,\tau} > 0. \tag{34}$$

$\alpha(i, \theta_1, \theta_2) > 0$ comes from the fact that the logistic sigmoid function is strictly increasing and has a gradient greater than zero at every point. Because of this inequality, each element of $\mathbf{G}_T$ denoted

by $[\mathbf{G}_T]_{lk} \forall l, k \in [d]$, is strictly larger than each element of $c_{\sigma,\tau}[\hat{\Sigma}]_{lk}$. We use this to prove that $\mathbf{G}_T \succeq c_{\sigma,\tau}(\hat{\Sigma} + \lambda I)$ for any $v = \theta_1 - \theta_2$. We first remind the reader of the definition of $\hat{\Sigma}$:

$$\hat{\Sigma} = \frac{1}{n} \sum_{i=1}^{n} \gamma^{T-t_i-1}(\phi(x_i, a_i) - \phi(x_i, a_i'))(\phi(x_i, a_i) - \phi(x_i, a_i'))^{\mathsf{T}}.$$

We then prove the inequality, using the fact that $\alpha$ and $\gamma$ do not depend upon the indices $l, k$ of the vector $v$ to move the sum across indices within the sum over the datapoints

$$v^{\mathsf{T}} \mathbf{G}_T v = \sum_{(l,k) \in [d]^2} \left[ \frac{1}{n} \sum_{i=1}^{n} \gamma^{T-1-t_i} \alpha(i, \theta_1, \theta_2) \hat{\phi}_i \hat{\phi}_i^{\mathsf{T}} + \lambda c_{\sigma,\tau} I \right]_{lk} v_l v_k$$

$$= \left( \frac{1}{n} \sum_{i=1}^{n} \gamma^{T-1-t_i} \alpha(i, \theta_1, \theta_2) \sum_{(l,k) \in [d]^2} \left[ \hat{\phi}_i \hat{\phi}_i^{\mathsf{T}} \right]_{lk} v_l v_k \right) + \lambda c_{\sigma,\tau} \sum_{l \in [d]} v_l^2$$

$$\geq \left( \frac{1}{n} \sum_{i=1}^{n} \gamma^{T-1-t_i} c_{\sigma,\tau} \sum_{(l,k) \in [d]^2} \left[ \hat{\phi}_i \hat{\phi}_i^{\mathsf{T}} \right]_{lk} v_l v_k \right) + \lambda c_{\sigma,\tau} \sum_{l \in [d]} v_l^2 \qquad (35)$$

$$= c_{\sigma,\tau} \sum_{(l,k) \in [d]^2} \underbrace{\left[ \frac{1}{n} \sum_{i=1}^{n} \gamma^{T-1-t_i} \hat{\phi}_i \hat{\phi}_i^{\mathsf{T}} + \lambda I \right]}_{\hat{\Sigma} + \lambda I}_{lk} v_l v_k = c_{\sigma,\tau} v^{\mathsf{T}}(\hat{\Sigma} + \lambda I) v. \qquad (36)$$

We now continue applying Equation (33) to bound the estimation error term:

$$\|\tilde{\theta}_T - \theta_T^*\|_{\hat{\Sigma}+\lambda I} = \frac{1}{\tau^2} \|\mathbf{G}_T^{-1}(g^\tau(\tilde{\theta}_T) - g^\tau(\theta_T^*))\|_{\hat{\Sigma}+\lambda I}. \qquad (37)$$

We use Equation (36) to apply $\mathbf{G}_T^{-1} \prec \frac{1}{c_{\sigma,\tau}}(\hat{\Sigma} + \lambda I)^{-1}$:

$$\frac{1}{\tau^2} \|\mathbf{G}_T^{-1}(g^\tau(\tilde{\theta}_T) - g^\tau(\theta_T^*))\|_{\hat{\Sigma}+\lambda I} \prec \frac{1}{\tau^2 c_{\sigma,\tau}} \|g^\tau(\tilde{\theta}_T) - g^\tau(\theta_T^*)\|_{(\hat{\Sigma}+\lambda I)^{-1}}. \qquad (38)$$

We add and subtract $g^\tau(\hat{\theta}_T)$ inside Equation (38), and apply triangle inequality to derive

$$\frac{1}{\tau^2 c_{\sigma,\tau}} \|g^\tau(\tilde{\theta}_T) - g^\tau(\theta_T^*)\|_{(\hat{\Sigma}+\lambda I)^{-1}}$$

$$= \frac{1}{\tau^2 c_{\sigma,\tau}} \|g^\tau(\tilde{\theta}_T) - g^\tau(\hat{\theta}_T) + g^\tau(\hat{\theta}_T) - g^\tau(\theta_T^*)\|_{(\hat{\Sigma}+\lambda I)^{-1}}$$

$$\leq \frac{1}{\tau^2 c_{\sigma,\tau}} \left( \|g^\tau(\tilde{\theta}_T) - g^\tau(\hat{\theta}_T)\|_{(\hat{\Sigma}+\lambda I)^{-1}} + \|g^\tau(\hat{\theta}_T) - g^\tau(\theta_T^*)\|_{(\hat{\Sigma}+\lambda I)^{-1}} \right). \qquad (39)$$

We use the definition of $\tilde{\theta}_T$ from Equation (28) to derive $\|g^\tau(\tilde{\theta}_T) - g^\tau(\hat{\theta}_T)\|_{(\hat{\Sigma}+\lambda I)^{-1}} \leq \|g^\tau(\hat{\theta}_T) - g^\tau(\theta_T^*)\|_{(\hat{\Sigma}+\lambda I)^{-1}}$ and get

$$\frac{1}{\tau^2 c_{\sigma,\tau}} \left( \|g^\tau(\tilde{\theta}_T) - g^\tau(\hat{\theta}_T)\|_{(\hat{\Sigma}+\lambda I)^{-1}} + \|g^\tau(\hat{\theta}_T) - g^\tau(\theta_T^*)\|_{(\hat{\Sigma}+\lambda I)^{-1}} \right)$$

$$\leq \frac{2}{\tau^2 c_{\sigma,\tau}} \|g^\tau(\hat{\theta}_T) - g^\tau(\theta_T^*)\|_{(\hat{\Sigma}+\lambda I)^{-1}}. \qquad (40)$$

We remind the definition of $\hat{\theta}_T$, which minimises the gradient of the loss defined in Equation (27), making $\nabla \mathcal{L}_{\text{reg}}^{\text{NS}}(\theta) = 0$:

$$\nabla \mathcal{L}_{\text{reg}}^{\text{NS}}(\theta) = \frac{1}{n} \sum_{i=1}^{n} \tau \gamma^{T-1-t_i} \left[ \sigma(\tau \langle \hat{\phi}_i, \hat{\theta}_T - \theta_{\text{ref}} \rangle) - o_i \right] \hat{\phi}_i + \lambda c_{\sigma,\tau} \tau^2 \hat{\theta}_T = 0. \qquad (41)$$

We rearrange the terms in Equation (41) to derive $g^\tau(\hat{\theta}_T)$ on one side of the equation:

$$\underbrace{\frac{1}{n}\sum_{i=1}^{n}\tau\gamma^{T-1-t_i}\sigma(\tau\langle\hat{\phi}_i,\hat{\theta}_T-\theta_{\text{ref}}\rangle)\hat{\phi}_i + \lambda c_{\sigma,\tau}\tau^2\hat{\theta}_T}_{=g^\tau(\hat{\theta}_T)} = \frac{1}{n}\sum_{i=1}^{n}\tau\gamma^{T-1-t_i}o_i\hat{\phi}_i. \tag{42}$$

We apply the result of Equation (42) to obtain

$$g^\tau(\hat{\theta}_T) - g^\tau(\theta_T^*) = \frac{1}{n}\sum_{i=1}^{n}\tau\gamma^{T-1-t_i}[o_i - \sigma(h_{\theta_T^*}(x_i,a_i,a_i'))]\hat{\phi}_i - \lambda c_{\sigma,\tau}\tau^2\theta_T^*. \tag{43}$$

Using the fact that the preference label $o_i$ is obtained from the optimal parameter at time step $t_i$, we define $\epsilon_i = o_i - \sigma(\tau\langle\hat{\phi}_i,\theta_{t_i}^*-\theta_{\text{ref}}\rangle)$, and use $o_i = \epsilon_i + \sigma(\tau\langle\hat{\phi}_i,\theta_{t_i}^*-\theta_{\text{ref}}\rangle)$ to get

$$\frac{1}{n}\sum_{i=1}^{n}\tau\gamma^{T-1-t_i}[o_i - \sigma(h_{\theta_T^*}(x_i,a_i,a_i'))]\hat{\phi}_i - \lambda c_{\sigma,\tau}\tau^2\theta_T^*$$

$$= \frac{1}{n}\sum_{i=1}^{n}\tau\gamma^{T-1-t_i}[\epsilon_i + \sigma(\tau\langle\hat{\phi}_i,\theta_{t_i}^*-\theta_{\text{ref}}\rangle) - \sigma(h_{\theta_T^*}(x_i,a_i,a_i'))]\hat{\phi}_i - \lambda c_{\sigma,\tau}\tau^2\theta_T^*$$

$$= \underbrace{\frac{1}{n}\sum_{i=1}^{n}\tau\gamma^{T-1-t_i}[\sigma(\tau\langle\hat{\phi}_i,\theta_{t_i}^*-\theta_{\text{ref}}\rangle) - \sigma(h_{\theta_T^*}(x_i,a_i,a_i'))]\hat{\phi}_i}_{\text{tracking}}$$

$$+ \underbrace{\frac{1}{n}\sum_{i=1}^{n}\tau\gamma^{T-1-t_i}\epsilon_i\hat{\phi}_i - \lambda c_{\sigma,\tau}\tau^2\theta_T^*}_{\text{learning}}. \tag{44}$$

We use terms in Equation (44) with Equation (40) to define learning error and tracking error:

$$\xi^{\text{learn}} = \frac{2}{\tau^2 c_{\sigma,\tau}}\|\frac{1}{n}\sum_{i=1}^{n}\tau\gamma^{T-1-t_i}\epsilon_i\hat{\phi}_i - \lambda c_{\sigma,\tau}\tau^2\theta_T^*\|_{(\hat{\Sigma}+\lambda I)^{-1}} \tag{45}$$

$$\xi^{\text{track}} = \frac{2}{\tau^2 c_{\sigma,\tau}}\|\frac{1}{n}\sum_{i=1}^{n}\tau\gamma^{T-1-t_i}[\sigma(\tau\langle\hat{\phi}_i,\theta_{t_i}^*-\theta_{\text{ref}}\rangle) - \sigma(h_{\theta_T^*}(x_i,a_i,a_i'))]\hat{\phi}_i\|_{(\hat{\Sigma}+\lambda I)^{-1}}. \tag{46}$$

Bounding each of Equation (45) and Equation (46) results in Theorem 1. The detailed bounds for the tracking and learning terms are provided in Appendix E.1.2 and Appendix E.1.3 respectively.

### E.1.2 CONFIDENCE SETS: LEARNING

We begin with the definition of the learning error:

$$\xi^{\text{learn}} = \frac{2}{\tau^2 c_{\sigma,\tau}}\|\frac{1}{n}\sum_{i=1}^{n}\tau\gamma^{T-1-t_i}\epsilon_i\hat{\phi}_i - \lambda c_{\sigma,\tau}\tau^2\theta_T^*\|_{(\hat{\Sigma}+\lambda I)^{-1}}. \tag{47}$$

We bound the norm of Equation (47) with respect to $\tilde{\Sigma} + \lambda I$, using the fact that $\hat{\Sigma} \succ \tilde{\Sigma}$ and $\tilde{\Sigma} + \lambda I \succeq \lambda I$:

$$\left\| \frac{1}{n} \sum_{i=1}^{n} \tau \gamma^{T-1-t_i} \epsilon_i \hat{\phi}_i - \lambda c_{\sigma,\tau} \tau^2 \theta_T^* \right\|_{(\hat{\Sigma}+\lambda I)^{-1}}$$

$$\leq \left\| \frac{1}{n} \sum_{i=1}^{n} \tau \gamma^{T-1-t_i} \epsilon_i \hat{\phi}_i - \lambda c_{\sigma,\tau} \tau^2 \theta_T^* \right\|_{(\tilde{\Sigma}+\lambda I)^{-1}}$$

$$\leq \left\| \lambda c_{\sigma,\tau} \tau^2 \theta_T^* \right\|_{(\lambda I)^{-1}} + \left\| \frac{1}{n} \sum_{i=1}^{n} \tau \gamma^{T-1-t_i} \epsilon_i \hat{\phi}_i \right\|_{(\tilde{\Sigma}+\lambda I)^{-1}}$$

$$\leq \tau^2 \sqrt{\lambda} c_{\sigma,\tau} W + \left\| \frac{1}{n} \sum_{i=1}^{n} \tau \gamma^{T-1-t_i} \epsilon_i \hat{\phi}_i \right\|_{(\tilde{\Sigma}+\lambda I)^{-1}}. \tag{48}$$

We can use the $\epsilon_i$'s property of being a sub-Gaussian random variable, sampled i.i.d. during the creation of the dataset. We apply Theorem 2.1 of (Hsu et al., 2012) to Equation (48), resulting in a bound holding with probability at least $1 - \delta$:

$$\left\| \frac{1}{n} \sum_{i=1}^{n} \tau \gamma^{T-1-t_i} \epsilon_i \hat{\phi}_i \right\|_{(\tilde{\Sigma}+\lambda I)^{-1}} \leq \tau C_1 \sqrt{\frac{d + \log(1/\delta)}{n}} = \beta_T(\delta), \tag{49}$$

where $C_1$ denotes a constant introduced for bounding purpose. We provide the details of applying (Hsu et al., 2012)'s theorem in Appendix E.4.

We now go back to the original definition of learning error term $\xi^{\text{learn}}$ and bound it. We use the result in Equation (48) and Equation (49) to derive

$$\xi^{\text{learn}} = \frac{2}{\tau^2 c_{\sigma,\tau}} \left\| \frac{1}{n} \sum_{i=1}^{n} \tau \gamma^{T-1-t_i} \epsilon_i \hat{\phi}_i - \lambda c_{\sigma,\tau} \tau^2 \theta_T^* \right\|_{(\hat{\Sigma}+\lambda I)^{-1}}$$

$$= \frac{2}{\tau^2 c_{\sigma,\tau}} \left( \tau^2 \sqrt{\lambda} c_{\sigma,\tau} W + \tau C_1 \sqrt{\frac{d + \log(1/\delta)}{n}} \right)$$

$$= 2\sqrt{\lambda} W + \frac{2 C_1}{\tau c_{\sigma,\tau}} \sqrt{\frac{d + \log(1/\delta)}{n}}, \tag{50}$$

which finishes the bounding of the learning error.

### E.1.3 ESTIMATION ERROR: TRACKING

We begin with the definition of the tracking error:

$$\xi^{\text{track}} = \frac{2}{\tau^2 c_{\sigma,\tau}} \left\| \frac{1}{n} \sum_{i=1}^{n} \tau \gamma^{T-1-t_i} [\sigma(\tau \langle \hat{\phi}_i, \theta_{t_i}^* - \theta_{\text{ref}} \rangle) - \sigma(h_{\theta_T^*}(x_i, a_i, a_i'))] \hat{\phi}_i \right\|_{(\hat{\Sigma}+\lambda I)^{-1}}$$

$$= \frac{2}{\tau^2 c_{\sigma,\tau}} \left\| \frac{1}{n} \sum_{i=1}^{n} \tau \gamma^{T-1-t_i} [\sigma(\tau \langle \hat{\phi}_i, \theta_{t_i}^* - \theta_{\text{ref}} \rangle) - \sigma(\tau \langle \hat{\phi}_i, \theta_T^* - \theta_{\text{ref}} \rangle)] \hat{\phi}_i \right\|_{(\hat{\Sigma}+\lambda I)^{-1}}. \tag{51}$$

We remind that using Equation (34), $\alpha(i, \theta_{t_i}^*, \theta_T^*)$ is

$$\alpha(i, \theta_{t_i}^*, \theta_T^*) := \int_{v=0}^{1} \dot{\sigma}(\tau \langle \hat{\phi}_i, (1-v)\theta_T^* + v\theta_{t_i}^* \rangle) dv. \tag{52}$$

Applying the man value theorem to Equation (51), we obtain

$$\frac{2}{\tau^2 c_{\sigma,\tau}} \left\| \frac{1}{n} \sum_{i=1}^{n} \tau \gamma^{T-1-t_i} [\sigma(\tau \langle \hat{\phi}_i, \theta_{t_i}^* - \theta_{\text{ref}} \rangle) - \sigma(\tau \langle \hat{\phi}_i, \theta_T^* - \theta_{\text{ref}} \rangle)] \hat{\phi}_i \right\|_{(\hat{\Sigma}+\lambda I)^{-1}}$$

$$= \frac{2}{\tau^2 c_{\sigma,\tau}} \left\| \frac{1}{n} \sum_{i=1}^{n} \tau^2 \gamma^{T-1-t_i} \alpha(i, \theta_{t_i}^*, \theta_T^*) \hat{\phi}_i \hat{\phi}_i^\mathsf{T} (\theta_{t_i}^* - \theta_T^*) \right\|_{(\hat{\Sigma}+\lambda I)^{-1}}. \tag{53}$$

We apply telescopic sum, which separates $\theta_{t_i}^* - \theta_T^*$ into differences of the optimal parameters between each datapoint:

$$\left\| \frac{1}{n} \sum_{i=1}^n \tau^2 \gamma^{T-1-t_i} \alpha(i, \theta_{t_i}^*, \theta_T^*) \hat{\phi}_i \hat{\phi}_i^\intercal (\theta_{t_i}^* - \theta_T^*) \right\|_{(\hat{\Sigma} + \lambda I)^{-1}}$$

$$= \left\| \frac{1}{n} \sum_{i=1}^n \tau^2 \gamma^{T-1-t_i} \alpha(i, \theta_{t_i}^*, \theta_T^*) \hat{\phi}_i \hat{\phi}_i^\intercal \left( \sum_{p=i}^n (\theta_{t_p}^* - \theta_{t_{p+1}}^*) \right) \right\|_{(\hat{\Sigma} + \lambda I)^{-1}}, \tag{54}$$

where we use $t_{n+1}$ to denote $T$.

Then we use $\sum_{i=k}^n \sum_{j=i}^n a_{i,j} = \sum_{j=k}^n \sum_{i=k}^j a_{i,j}$ to rearrange the terms inside the summation:

$$\left\| \frac{1}{n} \sum_{i=1}^n \tau^2 \gamma^{T-1-t_i} \alpha(i, \theta_{t_i}^*, \theta_T^*) \hat{\phi}_i \hat{\phi}_i^\intercal \left( \sum_{p=i}^n (\theta_{t_p}^* - \theta_{t_{p+1}}^*) \right) \right\|_{(\hat{\Sigma} + \lambda I)^{-1}}$$

$$= \left\| \sum_{p=1}^n \frac{1}{n} \sum_{i=1}^p \tau^2 \gamma^{T-1-t_i} \alpha(i, \theta_{t_i}^*, \theta_T^*) \hat{\phi}_i \hat{\phi}_i^\intercal (\theta_{t_p}^* - \theta_{t_{p+1}}^*) \right\|_{(\hat{\Sigma} + \lambda I)^{-1}}. \tag{55}$$

We use $\alpha(i, \theta_{t_i}^*, \theta_T^*) \le k_{\sigma,\tau}$ using the definition of $\alpha_i$ in Equation (34) to get

$$\left\| \sum_{p=1}^n \frac{1}{n} \sum_{i=1}^p \tau^2 \gamma^{T-1-t_i} \alpha(i, \theta_{t_i}^*, \theta_T^*) \hat{\phi}_i \hat{\phi}_i^\intercal (\theta_{t_p}^* - \theta_{t_{p+1}}^*) \right\|_{(\hat{\Sigma} + \lambda I)^{-1}}$$

$$\le \tau^2 k_{\sigma,\tau} \left\| \sum_{p=1}^n \frac{1}{n} \sum_{i=1}^p \gamma^{T-1-t_i} \hat{\phi}_i \hat{\phi}_i^\intercal (\theta_{t_p}^* - \theta_{t_{p+1}}^*) \right\|_{(\hat{\Sigma} + \lambda I)^{-1}}. \tag{56}$$

We then apply triangle inequality and Cauchy-Schwarz inequality to get

$$\tau^2 k_{\sigma,\tau} \left\| \sum_{p=1}^n \frac{1}{n} \sum_{i=1}^p \gamma^{T-1-t_i} \hat{\phi}_i \hat{\phi}_i^\intercal (\theta_{t_p}^* - \theta_{t_{p+1}}^*) \right\|_{(\hat{\Sigma} + \lambda I)^{-1}}$$

$$\le \tau^2 k_{\sigma,\tau} \sum_{p=1}^n \left\| \frac{1}{n} \sum_{i=1}^p \gamma^{T-1-t_i} \hat{\phi}_i \|\hat{\phi}_i^\intercal\|_2 \|\theta_{t_p}^* - \theta_{t_{p+1}}^*\|_2 \right\|_{(\hat{\Sigma} + \lambda I)^{-1}}. \tag{57}$$

We use $\|\hat{\phi}\| \le 2L$ and arrange terms to obtain

$$\tau^2 k_{\sigma,\tau} \sum_{p=1}^n \left\| \frac{1}{n} \sum_{i=1}^p \gamma^{T-1-t_i} \hat{\phi}_i \|\hat{\phi}_i^\intercal\|_2 \|\theta_{t_p}^* - \theta_{t_{p+1}}^*\|_2 \right\|_{(\hat{\Sigma} + \lambda I)^{-1}}$$

$$\le 2L\tau^2 k_{\sigma,\tau} \sum_{p=1}^n \underbrace{\frac{1}{n} \sum_{i=1}^p \gamma^{T-1-t_i} \|\hat{\phi}_i\|_{(\hat{\Sigma} + \lambda I)^{-1}}}_{=v_1} \|\theta_{t_p}^* - \theta_{t_{p+1}}^*\|_2. \tag{58}$$

Here we bound the term $v_1$. We first apply Jensen's inequality to derive

$$v_1 \le \sqrt{\frac{1}{n} \sum_{i=1}^p \gamma^{T-1-t_i}} \sqrt{\frac{1}{n} \sum_{i=1}^p \gamma^{T-1-t_i} \|\hat{\phi}_i\|_{(\hat{\Sigma} + \lambda I)^{-1}}^2}$$

$$= \gamma^{\frac{T-1}{2}} \sqrt{\frac{1}{n} \sum_{i=1}^p \gamma^{-t_i}} \sqrt{\frac{1}{n} \sum_{i=1}^p \gamma^{T-1-t_i} \|\hat{\phi}_i\|_{(\hat{\Sigma} + \lambda I)^{-1}}^2}. \tag{59}$$

We then use the property of trace operation and $\hat{\Sigma} \succ \sum_{i=1}^{p} \gamma^{T-1-t_i} \hat{\phi}_i \hat{\phi}_i^{\mathsf{T}}$ from Equation (17) to get

$$
\begin{aligned}
\frac{1}{n} \sum_{i=1}^{p} \gamma^{T-1-t_i} \|\hat{\phi}_i\|_{(\hat{\Sigma}+\lambda I)^{-1}}^2 &= \frac{1}{n} \sum_{i=1}^{p} \gamma^{T-1-t_i} \mathrm{tr}\left(\hat{\phi}_i^{\mathsf{T}} (\hat{\Sigma}+\lambda I)^{-1} \hat{\phi}_i\right) \\
&= \mathrm{tr}\left((\hat{\Sigma}+\lambda I)^{-1} \frac{1}{n} \sum_{i=1}^{p} \gamma^{T-1-t_i} \hat{\phi}_i \hat{\phi}_i^{\mathsf{T}}\right) \\
&\leq \mathrm{tr}\left(I_d\right) = d.
\end{aligned}
\tag{60}
$$

We apply Assumption 5 here. Because each time step can have at maximum $\bar{m}$ datapoints, we can upper bound $\frac{1}{n} \sum_{i=1}^{p} \gamma^{-t_i}$ with

$$
\frac{1}{n} \sum_{i=1}^{p} \gamma^{-t_i} \leq \frac{\bar{m}}{n} \sum_{k=1}^{t} \gamma^{-k} = \frac{\bar{m}\gamma(\gamma^{-(t+1)}-1)}{n(1-\gamma)},
\tag{61}
$$

where $t = \left\lceil \frac{|[p]|}{\bar{m}} \right\rceil$. We combine Equation (60) and Equation (61) to obtain

$$
\begin{aligned}
2L\tau^2 k_{\sigma,\tau} \sum_{p=1}^{n} \frac{1}{n} \sum_{i=1}^{p} \gamma^{T-1-t_i} &\|\hat{\phi}_i\|_{(\hat{\Sigma}+\lambda I)^{-1}} \|\theta_{t_p}^* - \theta_{t_{p+1}}^*\|_2 \\
&\leq 2L\tau^2 k_{\sigma,\tau} \sum_{p=1}^{n} \gamma^{\frac{T-1}{2}} \sqrt{\frac{d\bar{m}\gamma(\gamma^{-(t+1)}-1)}{n(1-\gamma)}} \|\theta_{t_p}^* - \theta_{t_{p+1}}^*\|_2.
\end{aligned}
\tag{62}
$$

We apply Assumption 5 again to upper bound the summation as $\sum_{p=1}^{n} v_p \leq \bar{m} \sum_{t=1}^{T-1} v_t$, getting

$$
\begin{aligned}
2L\tau^2 k_{\sigma,\tau} \sum_{p=1}^{n} \gamma^{\frac{T-1}{2}} &\sqrt{\frac{d\bar{m}\gamma(\gamma^{-(t+1)}-1)}{n(1-\gamma)}} \|\theta_{t_p}^* - \theta_{t_{p+1}}^*\|_2 \\
&\leq 2L\tau^2 k_{\sigma,\tau} \bar{m} \sum_{t=1}^{T-1} \gamma^{\frac{T-1}{2}} \sqrt{\frac{d\bar{m}\gamma(\gamma^{-(t+1)}-1)}{n(1-\gamma)}} \|\theta_t^* - \theta_{t+1}^*\|_2.
\end{aligned}
\tag{63}
$$

We apply $v = \frac{1}{T} \sum_{k=1}^{T} v$ to introduce another summation:

$$
\begin{aligned}
2L\tau^2 k_{\sigma,\tau} \bar{m} \sum_{t=1}^{T-1} \gamma^{\frac{T-1}{2}} &\sqrt{\frac{d\bar{m}\gamma(\gamma^{-(t+1)}-1)}{n(1-\gamma)}} \|\theta_t^* - \theta_{t+1}^*\|_2 \\
&= \frac{2L\tau^2 k_{\sigma,\tau} \bar{m}}{T} \sum_{k=1}^{T} \sum_{t=1}^{T-1} \gamma^{\frac{T-1}{2}} \sqrt{\frac{d\bar{m}\gamma(\gamma^{-(t+1)}-1)}{n(1-\gamma)}} \|\theta_t^* - \theta_{t+1}^*\|_2.
\end{aligned}
\tag{64}
$$

Because $\gamma < 1$, we can bound

$$
\sum_{k=1}^{T} \sum_{t=1}^{T-1} \gamma^{\frac{T-1}{2}} \sqrt{\frac{d\bar{m}\gamma(\gamma^{-(t+1)}-1)}{n(1-\gamma)}} \leq 2 \sum_{t=1}^{T-1} \sum_{k=t+1}^{T} \gamma^{\frac{k-1}{2}} \sqrt{\frac{d\bar{m}\gamma(\gamma^{-(t+1)}-1)}{n(1-\gamma)}}
\tag{65}
$$

and apply geometric sum to obtain

$$
2 \sum_{t=1}^{T-1} \sum_{k=t+1}^{T} \gamma^{\frac{k-1}{2}} \sqrt{\frac{d\bar{m}\gamma(\gamma^{-(t+1)}-1)}{n(1-\gamma)}} = 2 \sum_{t=1}^{T-1} \frac{\gamma^{\frac{t}{2}} - \gamma^{\frac{T}{2}}}{1-\gamma^{\frac{1}{2}}} \sqrt{\frac{d\bar{m}\gamma(\gamma^{-(t+1)}-1)}{n(1-\gamma)}}.
\tag{66}
$$

We use $\gamma < 1$ again to derive $\frac{1+\gamma^{\frac{1}{2}}}{2} < 1$, and get

$$
\begin{aligned}
2 \sum_{t=1}^{T-1} \frac{\gamma^{\frac{t}{2}} - \gamma^{\frac{T}{2}}}{1-\gamma^{\frac{1}{2}}} \sqrt{\frac{d\bar{m}\gamma(\gamma^{-(t+1)}-1)}{n(1-\gamma)}} &\leq 2 \sum_{t=1}^{T-1} \frac{\gamma^{\frac{t}{2}} - \gamma^{\frac{T}{2}}}{1-\gamma^{\frac{1}{2}} \frac{1+\gamma^{\frac{1}{2}}}{2}} \sqrt{\frac{d\bar{m}\gamma(\gamma^{-(t+1)}-1)}{n(1-\gamma)}} \\
&= 4 \sum_{t=1}^{T-1} \frac{\gamma^{\frac{t}{2}} - \gamma^{\frac{T}{2}}}{1-\gamma} \sqrt{\frac{d\bar{m}\gamma(\gamma^{-(t+1)}-1)}{n(1-\gamma)}}.
\end{aligned}
\tag{67}
$$

We then use $\left(\gamma^{\frac{t}{2}} - \gamma^{\frac{T}{2}}\right)\sqrt{\gamma(\gamma^{-(t+1)}-1)} \le \gamma^{\frac{t}{2}}\gamma^{-\frac{t}{2}} = 1$ to derive

$$4\sum_{t=1}^{T-1}\frac{\gamma^{\frac{t}{2}} - \gamma^{\frac{T}{2}}}{1-\gamma}\sqrt{\frac{d\bar{m}\gamma(\gamma^{-(t+1)}-1)}{n(1-\gamma)}} \le 4\sqrt{\frac{d\bar{m}}{n}}\sum_{t=1}^{T-1}\frac{1}{(1-\gamma)^{\frac{3}{2}}}. \tag{68}$$

We use the result from Equation (68) to Equation (64), and use the definition of variation budget $B_T$ from Assumption 3 to get

$$\frac{2L\tau^2 k_{\sigma,\tau}\bar{m}}{T}\sum_{k=1}^{T}\sum_{t=1}^{T-1}\gamma^{\frac{T-1}{2}}\sqrt{\frac{d\bar{m}\gamma(\gamma^{-(t+1)}-1)}{n(1-\gamma)}}\|\theta_t^* - \theta_{t+1}^*\|_2$$

$$\le \frac{8L\tau^2 k_{\sigma,\tau}\bar{m}}{T}\sqrt{\frac{d\bar{m}}{n}}\sum_{t=1}^{T-1}\frac{1}{(1-\gamma)^{\frac{3}{2}}}\|\theta_t^* - \theta_{t+1}^*\|_2$$

$$\le \frac{8L\tau^2 k_{\sigma,\tau}\bar{m}}{T(1-\gamma)^{\frac{3}{2}}}\sqrt{\frac{d\bar{m}}{n}}B_T. \tag{69}$$

We now combine Equation (69) with Equation (46) to derive the full bound of the tracking error:

$$\xi^{\text{track}} = \frac{16LR_{\sigma,\tau}\bar{m}}{T(1-\gamma)^{\frac{3}{2}}}\sqrt{\frac{d\bar{m}}{n}}B_T. \tag{70}$$

We now use Equation (70) with Equation (50) to obtain the full estimation error:

$$\|\hat{\theta}_T - \hat{\theta}_T^*\|_{\hat{\Sigma}+\lambda I} \le \xi^{\text{learn}} + \xi^{\text{track}}$$

$$\le 2\sqrt{\lambda}W + \frac{2C_1}{\tau c_{\sigma,\tau}}\sqrt{\frac{d+\log(1/\delta)}{n}} + \frac{16LR_{\sigma,\tau}\bar{m}}{T(1-\gamma)^{\frac{3}{2}}}\sqrt{\frac{d\bar{m}}{n}}B_T, \tag{71}$$

which concludes the analysis for Theorem 1.

## E.2    REGRET BOUND

**Theorem 2.** *(Regret bound of $\tilde{\theta}_T$) Let $\delta \in (0, \frac{1}{2}], \tau > 0$. Let $\tilde{\theta}_T$ denote the parameter in $\Theta$ which minimises the NS-DPO loss (Equation (12)) on an offline dataset. The following bound holds with probability at least $1 - 2\delta$ and when $\lambda \ge C\sqrt{d\log(4d/\delta)/n}$:*

$$R_T^{\text{off}} \le \frac{\tau\kappa\bar{m}T(1-\gamma)}{2\underline{m}(1-\gamma^{T-1})}\left(2\sqrt{\lambda}W + \frac{2C_1}{\tau c_{\sigma,\tau}}\sqrt{\frac{d+\log(1/\delta)}{n}} + \frac{16LR_{\sigma,\tau}\bar{m}}{T(1-\gamma)^{\frac{3}{2}}}\sqrt{\frac{d\bar{m}}{n}}B_T\right)^2,$$

*where $C_1 > 0$ denotes a constant. When $\gamma = 1 - \left(\frac{B_T}{T}\right)^{3/4}$, $R_T^{\text{off}}$ satisfies:*

$$R_T^{\text{off}} = \tilde{O}\left(d\,B_T^{3/4}\,n^{-1/4}\right).$$

### E.2.1    POPULATION COVARIANCE OF FEATURE DIFFERENCES

Let $\Sigma_{\pi_{\text{ref}},\text{diff}}$ define the population covariance matrix of the feature differences:

$$\Sigma_{\pi_{\text{ref}},\text{diff}} = \mathbb{E}[\hat{\phi}\hat{\phi}^{\mathsf{T}}], \tag{72}$$

where $\hat{\phi} = \phi(x,a) - \phi(x,a')$ denotes the feature difference vector, and the expectation is computed with respect to $x \sim \mathcal{X}, t \sim \mathcal{T}, a, a' \sim \pi_{\text{ref}}(\cdot|x)$. We also define the discounted population covariance matrix $\Sigma_{\pi_{\text{ref}},\text{diff}}^{\gamma}$:

$$\Sigma_{\pi_{\text{ref}},\text{diff}}^{\gamma} = \mathbb{E}[\gamma^{T-1-t}\hat{\phi}\hat{\phi}^{\mathsf{T}}], \tag{73}$$

where the expectation is computed with respect to the same distributions as $\Sigma_{\pi_{\text{ref}},\text{diff}}$.

We then define $\omega^{\mathrm{upp}}(T, \gamma)$:

$$\omega^{\mathrm{upp}}(T, \gamma) = \sup_{v \in \mathbb{R}^d} \frac{v^\mathsf{T} \Sigma_{\pi_{\mathrm{ref}}, \mathrm{diff}} v}{v^\mathsf{T} \Sigma^\gamma_{\pi_{\mathrm{ref}}, \mathrm{diff}} v}, \tag{74}$$

Without any assumptions on the time distribution, $\omega^{\mathrm{upp}}(T, \gamma) \leq \gamma^{-(T-1)}$, which happens when all the datapoints come from the oldest time step. We use Assumption 5 to obtain a tighter upper bound of $\omega^{\mathrm{upp}}$. Using $\underline{m}(T-1) \leq n \leq \bar{m}(T-1)$, we can get

$$\frac{1}{n} \sum_{i=1}^n \gamma^{T-1-t_i} \geq \frac{\underline{m}}{n} \cdot \sum_{t=1}^{T-1} \gamma^{T-1-t} \geq \frac{\underline{m}}{\bar{m}(T-1)} \cdot \sum_{t=1}^{T-1} \gamma^{T-1-t}. \tag{75}$$

We note that the prompt distribution $\mathcal{X}$ and the reference policy $\pi_{\mathrm{ref}}$ are independent from the time step distribution $\mathcal{T}$. Using Equation (75), we obtain

$$v^\mathsf{T} \Sigma^\gamma_{\pi_{\mathrm{ref}}, \mathrm{diff}} v \geq \left( \frac{\underline{m}}{\bar{m}(T-1)} \sum_{i=0}^{T-2} \gamma^i \right) \cdot (v^\mathsf{T} \Sigma_{\pi_{\mathrm{ref}}, \mathrm{diff}} v) = \frac{\underline{m}(1 - \gamma^{T-1})}{\bar{m}(T-1)(1-\gamma)} \cdot (v^\mathsf{T} \Sigma_{\pi_{\mathrm{ref}}, \mathrm{diff}} v), \tag{76}$$

which implies $\omega^{\mathrm{upp}}(T, \gamma) \leq \frac{\bar{m}(T-1)(1-\gamma)}{\underline{m}(1-\gamma^{T-1})}$.

### E.2.2 Decomposing Regret Bound

In order to decompose and bound the detailed elements of the regret bound, we first show the relation between the regret and the estimation error of the model parameters.

**Theorem 4.** *Let $\delta \in [0, 1]$. Let $\tilde{\theta}_T$ denote the parameter obtained by performing the parameter projection in Appendix E, after training with the NS-DPO loss defined in Equation (12) on an offline dataset. When $\lambda \geq C \sqrt{d \log(4d/\delta)/n}$, with probability at least $1 - \delta$:*

$$R_T^{\mathrm{off}} \leq \frac{\tau \kappa \bar{m} T (1 - \gamma)}{2 \underline{m}(1 - \gamma^{T-1})} \| \theta_T^* - \tilde{\theta}_T \|_{\hat{\Sigma} + \lambda I}^2. \tag{77}$$

Let $\pi_{\tilde{\theta}_T}$ denote the policy we obtained by training with NS-DPO and performing parameter projection. We use $\Sigma_{\pi_{\tilde{\theta}_T}}$ to denote the population covariance matrix, whose expectation taken with respect to $\pi_{\tilde{\theta}_T}$. We assess the performance of $\pi_{\tilde{\theta}}$ using the difference in expected non-stationary RLHF objective $\mathcal{J}_T(\pi)$ defined in Equation (7), which is

$$\begin{aligned} \mathcal{J}_T(\pi) &= \mathbb{E}_{x \sim \mathcal{X}, a \sim \pi} \Big[ r(x, a, T) - \tau \mathrm{D}_{\mathrm{KL}}[\pi(\cdot|x) \| \pi_{\mathrm{ref}}(\cdot|x)] \Big], \\ R_T^{\mathrm{off}} &= \mathcal{J}_T(\pi_T^*) - \mathcal{J}_T(\pi_{\tilde{\theta}_T}) \\ &= \mathbb{E}_{x \sim \mathcal{X}} \Big[ \mathbb{E}_{a \sim \pi_T^*(\cdot|x)}[r(x, a, T)] - \tau \mathrm{D}_{\mathrm{KL}}[\pi_T^*(\cdot|x) \| \pi_{\mathrm{ref}}(\cdot|x)] \\ &\qquad - \mathbb{E}_{a' \sim \pi_{\tilde{\theta}_T}(\cdot|x)}[r(x, a', T)] + \tau \mathrm{D}_{\mathrm{KL}}[\pi_{\tilde{\theta}_T}(\cdot|x) \| \pi_{\mathrm{ref}}(\cdot|x)] \Big]. \end{aligned} \tag{78}$$

We plug Equation (8) in Equation (78) to obtain

$$\begin{aligned} R_T^{\mathrm{off}} = \mathbb{E}_{x \sim \mathcal{X}} \Big[ \mathbb{E}_{a \sim \pi_T^*(\cdot|x)}[\tau \log \frac{\pi_T^*(a|x)}{\pi_{\mathrm{ref}}(a|x)}] - \tau \mathrm{D}_{\mathrm{KL}}[\pi_T^*(\cdot|x) \| \pi_{\mathrm{ref}}(\cdot|x)] \\ - \mathbb{E}_{a' \sim \pi_{\tilde{\theta}_T}(\cdot|x)}[\tau \log \frac{\pi_T^*(a|x)}{\pi_{\mathrm{ref}}(a|x)}] + \tau \mathrm{D}_{\mathrm{KL}}[\pi_{\tilde{\theta}_T}(\cdot|x) \| \pi_{\mathrm{ref}}(\cdot|x)] \Big], \end{aligned} \tag{79}$$

where terms with normalisation constant $Z_T^*(x)$ are cancelled out. By using the definition of KL divergence in Equation (79) again, we obtain

$$
\begin{aligned}
R_T^{\text{off}} &= \mathbb{E}_{x\sim\mathcal{X}}\left[-\mathbb{E}_{a'\sim\pi_{\tilde{\theta}_T}(\cdot|x)}\left[\tau\log\frac{\pi_T^*(a|x)}{\pi_{\text{ref}}(a|x)}\right]+\mathbb{E}_{a'\sim\pi_{\tilde{\theta}_T}(\cdot|x)}\left[\tau\log\frac{\pi_{\tilde{\theta}_T}(a|x)}{\pi_{\text{ref}}(a|x)}\right]\right] \\
&= \mathbb{E}_{x\sim\mathcal{X}}\left[\tau\mathbb{E}_{a'\sim\pi_{\tilde{\theta}_T}(\cdot|x)}\left[\log\frac{\pi_{\tilde{\theta}_T}(a|x)}{\pi_T^*(a|x)}\right]\right] \\
&= \mathbb{E}_{x\sim\mathcal{X}}\left[\tau D_{\text{KL}}[\pi_{\tilde{\theta}_T}(\cdot|x)\|\pi_T^*(\cdot|x)]\right].
\end{aligned}
\tag{80}
$$

Here, we borrow the analysis in Appendix A.5. of Chowdhury et al. (2024). We use the property of the Bergman divergence $\mathbb{B}_{\mathcal{L}_x}$ with its potential function $\mathcal{L}_x(\theta)=\log\sum_{a'\in\mathcal{A}}\langle\theta,\phi(x,a')\rangle$:

$$
D_{\text{KL}}[\pi_{\tilde{\theta}_T}(\cdot|x)\|\pi_T^*(\cdot|x)] = \frac{1}{2}(\theta_T^*-\tilde{\theta}_T)^{\mathsf{T}}\nabla^2\mathcal{L}_x(\theta)(\theta_T^*-\tilde{\theta}_T)
\tag{81}
$$

for a parameter $\theta\in\{t\tilde{\theta}+(1-t)\theta^* : t\in[0,1]\}$ using Taylor's approximation. With log-linear policies, $\mathbb{E}_{x\sim\mathcal{X}}[\nabla^2\mathcal{L}_x(\theta)]=\Sigma_{\pi_\theta}$. We use this to derive the upper bound of Equation (80):

$$
\begin{aligned}
R_T^{\text{off}} &= \mathbb{E}_{x\sim\mathcal{X}}\left[\tau D_{\text{KL}}[\pi_{\tilde{\theta}_T}(\cdot|x)\|\pi_T^*(\cdot|x)]\right] \\
&\leq \tau\|\theta_T^*-\tilde{\theta}_T\|_{\Sigma_{\pi_\theta}}^2 \\
&= \tau\|\theta_T^*-\tilde{\theta}_T\|_{\hat{\Sigma}+\lambda I}^2\frac{(\theta_T^*-\tilde{\theta}_T)^{\mathsf{T}}\Sigma_{\pi_\theta}(\theta_T^*-\tilde{\theta}_T)}{(\theta_T^*-\tilde{\theta}_T)^{\mathsf{T}}(\hat{\Sigma}+\lambda I)(\theta_T^*-\tilde{\theta}_T)}
\end{aligned}
\tag{82}
$$

We now use the following lemma from (Chowdhury et al., 2024), which relies on the matrix concentration inequality to explain the difference between $\hat{\Sigma}$ and $\Sigma_{\pi_{\text{ref}},\text{diff}}^\gamma$.

**Lemma 5.** *(Lemma A.1. of (Chowdhury et al., 2024)) With probability at least $1-\delta$, for some universal constant C, we have*

$$
\|\hat{\Sigma}-\Sigma_{\pi_{\text{ref}},\text{diff}}^\gamma\|_2 \leq C\sqrt{d\log(4d/\delta)/n}.
\tag{83}
$$

Lemma 5 implies that with probability at least $1-\delta$ and $\lambda\geq C\sqrt{d\log(4d/\delta)/n}$:

$$
\begin{aligned}
\hat{\Sigma}+\lambda I &\succeq \Sigma_{\pi_{\text{ref}},\text{diff}}^\gamma+\lambda I-C\sqrt{d\log(4d/\delta)/n} \\
&\succeq \Sigma_{\pi_{\text{ref}},\text{diff}}^\gamma.
\end{aligned}
\tag{84}
$$

We use Equation (84) to derive

$$
\begin{aligned}
&\tau\|\theta_T^*-\tilde{\theta}_T\|_{\hat{\Sigma}+\lambda I}^2\frac{(\theta_T^*-\tilde{\theta}_T)^{\mathsf{T}}\Sigma_{\pi_\theta}(\theta_T^*-\tilde{\theta}_T)}{(\theta_T^*-\tilde{\theta}_T)^{\mathsf{T}}(\hat{\Sigma}+\lambda I)(\theta_T^*-\tilde{\theta}_T)} \\
&\qquad\qquad\leq \tau\|\theta_T^*-\tilde{\theta}_T\|_{\hat{\Sigma}+\lambda I}^2\frac{(\theta_T^*-\tilde{\theta}_T)^{\mathsf{T}}\Sigma_{\pi_\theta}(\theta_T^*-\tilde{\theta}_T)}{(\theta_T^*-\tilde{\theta}_T)^{\mathsf{T}}\Sigma_{\pi_{\text{ref}},\text{diff}}^\gamma(\theta_T^*-\tilde{\theta}_T)}.
\end{aligned}
\tag{85}
$$

We then apply the result from Equation (74) which implies $(\|v\|_{\Sigma_{\pi_{\text{ref}},\text{diff}}^\gamma})^{-1} \leq \sqrt{\omega^{\text{upp}}(T,\gamma)}(\|v\|_{\Sigma_{\pi_{\text{ref}},\text{diff}}})^{-1}$:

$$
\begin{aligned}
&\tau\|\theta_T^*-\tilde{\theta}_T\|_{\hat{\Sigma}+\lambda I}^2\frac{(\theta_T^*-\tilde{\theta}_T)^{\mathsf{T}}\Sigma_{\pi_\theta}(\theta_T^*-\tilde{\theta}_T)}{(\theta_T^*-\tilde{\theta}_T)^{\mathsf{T}}\Sigma_{\pi_{\text{ref}},\text{diff}}^\gamma(\theta_T^*-\tilde{\theta}_T)} \\
&\qquad\qquad\leq \tau\omega^{\text{upp}}(T,\gamma)\|\theta_T^*-\tilde{\theta}_T\|_{\hat{\Sigma}+\lambda I}^2\frac{(\theta_T^*-\tilde{\theta}_T)^{\mathsf{T}}\Sigma_{\pi_\theta}(\theta_T^*-\tilde{\theta}_T)}{(\theta_T^*-\tilde{\theta}_T)^{\mathsf{T}}\Sigma_{\pi_{\text{ref}},\text{diff}}(\theta_T^*-\tilde{\theta}_T)}.
\end{aligned}
\tag{86}
$$

From the definition of $\Sigma_{\pi_{\mathrm{ref}},\mathrm{diff}}$ in Equation (72), $a, a'$ are independently sampled. We combine this fact with the population covariance matrix $\Sigma_{\pi_{\mathrm{ref}}}$, deriving $\Sigma_{\pi_{\mathrm{ref}},\mathrm{diff}} = 2\Sigma_{\pi_{\mathrm{ref}}}$. We use this to get

$$
\tau\omega^{\mathrm{upp}}(T,\gamma)\|\theta_T^* - \tilde{\theta}_T\|_{\hat{\Sigma}+\lambda I}^2 \frac{(\theta_T^* - \tilde{\theta}_T)^{\mathsf{T}}\Sigma_{\pi_\theta}(\theta_T^* - \tilde{\theta}_T)}{(\theta_T^* - \tilde{\theta}_T)^{\mathsf{T}}\Sigma_{\pi_{\mathrm{ref}},\mathrm{diff}}(\theta_T^* - \tilde{\theta}_T)}
$$

$$
= \frac{\tau\omega^{\mathrm{upp}}(T,\gamma)}{2}\|\theta_T^* - \tilde{\theta}_T\|_{\hat{\Sigma}+\lambda I}^2 \frac{(\theta_T^* - \tilde{\theta}_T)^{\mathsf{T}}\Sigma_{\pi_\theta}(\theta_T^* - \tilde{\theta}_T)}{(\theta_T^* - \tilde{\theta}_T)^{\mathsf{T}}\Sigma_{\pi_{\mathrm{ref}}}(\theta_T^* - \tilde{\theta}_T)}. \quad (87)
$$

We use $\kappa = \max_{\pi\in\Pi}\kappa_\pi$ with the definition of $\kappa_\pi$ in Equation (16), along with the result obtained in Equation (76) to use $\omega^{\mathrm{upp}}(T,\gamma) = \frac{(T-1)(1-\gamma)}{1-\gamma^{T-1}} \le \frac{T(1-\gamma)}{1-\gamma^{T-1}}$:

$$
\frac{\tau\omega^{\mathrm{upp}}(T,\gamma)}{2}\|\theta_T^* - \tilde{\theta}_T\|_{\hat{\Sigma}+\lambda I}^2 \frac{(\theta_T^* - \tilde{\theta}_T)^{\mathsf{T}}\Sigma_{\pi_\theta}(\theta_T^* - \tilde{\theta}_T)}{(\theta_T^* - \tilde{\theta}_T)^{\mathsf{T}}\Sigma_{\pi_{\mathrm{ref}}}(\theta_T^* - \tilde{\theta}_T)}
$$

$$
\le \frac{\tau\kappa\omega^{\mathrm{upp}}(T,\gamma)}{2}\|\theta_T^* - \tilde{\theta}_T\|_{\hat{\Sigma}+\lambda I}^2
$$

$$
\le \frac{\tau\kappa\bar{m}T(1-\gamma)}{2\underline{m}(1-\gamma^{T-1})}\|\theta_T^* - \tilde{\theta}_T\|_{\hat{\Sigma}+\lambda I}^2. \quad (88)
$$

### E.2.3 COMPLEXITY ANALYSIS

In order to investigate the complexity of the regret bound, we set the value of $\gamma$ using $T, B_T$. We first set $\gamma$ as

$$
\gamma = 1 - \left(\frac{B_T}{T}\right)^{3/4}. \quad (89)
$$

We apply Equation (89) in the estimation error $\|\theta_T^* - \tilde{\theta}_T\|_{\hat{\Sigma}+\lambda I}$, with assumption of $\lambda \ge C\sqrt{d\log(4d/\delta)/n}$ from Lemma 5, while ignoring the logarithmic factor:

$$
2\sqrt{\lambda}W \qquad\qquad (= d^{\frac{1}{4}}\, n^{-\frac{1}{4}})
$$

$$
\frac{2C_1}{\tau c_{\sigma,\tau}}\sqrt{\frac{d + \log(1/\delta)}{n}} \qquad\qquad (= d^{\frac{1}{2}}\, n^{-\frac{1}{2}})
$$

$$
\frac{16LR_{\sigma,\tau}\bar{m}}{T(1-\gamma)^{\frac{3}{2}}}\sqrt{\frac{d\bar{m}}{n}}B_T \qquad\qquad (= d^{\frac{1}{2}}\, B_T^{-\frac{1}{8}}\, T^{-\frac{3}{8}}) \quad (90)
$$

Here, we note that from Assumption 5, $n = \Theta(T)$. This allows us to consider the complexity with respect to the dataset size $n$ and $T$ together. We can conclude from Equation (90) that the complexity bound of the entire estimation error is $O(d^{\frac{1}{2}}\, T^{-\frac{1}{4}})$. By setting the value of $T$ to a sufficiently large one, making $1 - \gamma^{T-1} \ge \frac{1}{2}$, then the complexity of $\omega^{\mathrm{upp}}(T,\gamma)$ is

$$
T(1-\gamma) \qquad\qquad (= B_T^{\frac{3}{4}}\, T^{\frac{1}{4}}). \quad (91)
$$

Finally we present the total complexity bound of the algorithm, by applying the complexity of $\omega^{\mathrm{upp}}(T,\gamma)$ in Equation (91) to the squared estimation error $\|\theta_T^* - \tilde{\theta}_T\|_{\hat{\Sigma}+\lambda I}^2$:

$$
R_T^{\mathrm{off}} = O(d\, B_T^{\frac{3}{4}}\, T^{-\frac{1}{4}})
$$

$$
= O(d\, B_T^{\frac{3}{4}}\, n^{-\frac{1}{4}}). \quad (92)
$$

### E.3 THEORETICAL ANALYSIS OF NS-DPO UNDER STATIONARY PREFERENCES

**Corollary 3.** *(Regret bound under stationary preferences) Let $B_T \to 0$, $\delta \in (0, \frac{1}{2}], \tau > 0$. Let $\tilde{\theta}_T \in \Theta$ denote the minimiser of the NS-DPO loss (Equation (12)). Then, for $\lambda \geq C\sqrt{d \log(4d/\delta)/n}$, some constant $C_1 > 0$, $\gamma = 1 - \left(\frac{B_T}{T}\right)^\alpha$ and $\alpha \in (0, \frac{2}{3})$, we have with probability at least $1 - 2\delta$:*

$$\lim_{B_T \to 0} R_T^{\text{off}} < \underbrace{\frac{4\tau\kappa\bar{m}}{\underline{m}}}_{\text{Pre-factor}} \left( \sqrt{\lambda}W + \frac{C_1}{\tau c_{\sigma,\tau}} \sqrt{\frac{d + \log(1/\delta)}{n}} \right)^2,$$

*and recover the complexity of $R_T^{\text{off}} = O(n^{-\frac{1}{2}})$ under stationary preferences.*

We show that under certain conditions, NS-DPO's regret bound recovers $O(n^{-\frac{1}{2}})$. We first analyse the estimation error in the limit $B_T \to 0$. Consider the estimation error bound in Theorem 1:

$$\|\tilde{\theta}_T - \theta_T^*\|_{\hat{\Sigma}+\lambda I} \leq \underbrace{2\sqrt{\lambda}W + \frac{2C_1}{\tau c_{\sigma,\tau}} \sqrt{\frac{d + \log(1/\delta)}{n}}}_{\text{learning}} + \underbrace{\frac{16LR_{\sigma,\tau}\bar{m}}{T(1-\gamma)^{\frac{3}{2}}} \sqrt{\frac{d\bar{m}}{n}} B_T}_{\text{tracking}}, \tag{93}$$

in which the *tracking* term depends upon $\gamma$ and $B_T$. In the regret bound, we write $\gamma$ in terms of $B_T$ the form of

$$\gamma = 1 - \left(\frac{B_T}{T}\right)^\alpha, \tag{94}$$

where $\alpha \in \mathbb{R}$. We obtain $1 - \gamma = \left(\frac{B_T}{T}\right)^\alpha$ by rearranging terms. Substituting $B_T$ back into the estimation error bound, we find that the tracking term reduces to $16LR_{\sigma,\tau}\bar{m}T^{\frac{3}{2}\alpha-1}B_T^{1-\frac{3}{2}\alpha}\sqrt{\frac{d\bar{m}}{n}}$. By inspection, for $0 < \alpha < \frac{2}{3}$ the tracking term tends to 0 as $B_T \to 0$. Thus we conclude that

$$\lim_{B_T \to 0} \left( 2\sqrt{\lambda}W + \frac{2C_1}{\tau c_{\sigma,\tau}} \sqrt{\frac{d + \log(1/\delta)}{n}} + \frac{16LR_{\sigma,\tau}\bar{m}}{T(1-\gamma)^{\frac{3}{2}}} \sqrt{\frac{d\bar{m}}{n}} B_T \right) = 2\sqrt{\lambda}W + \frac{2C_1}{\tau c_{\sigma,\tau}} \sqrt{\frac{d + \log(1/\delta)}{n}}. \tag{95}$$

We now consider the regret bound in Theorem 2:

$$R_T^{\text{off}} \leq \underbrace{\frac{\tau\kappa\bar{m}T(1-\gamma)}{2\underline{m}(1-\gamma^{T-1})}}_{\text{Pre-factor}} \left( 2\sqrt{\lambda}W + \frac{2C_1}{\tau c_{\sigma,\tau}} \sqrt{\frac{d + \log(1/\delta)}{n}} + \underbrace{\frac{16LR_{\sigma,\tau}\bar{m}}{T(1-\gamma)^{\frac{3}{2}}} \sqrt{\frac{d\bar{m}}{n}} B_T}_{\text{Tracking}} \right)^2. \tag{96}$$

Here we note that the tracking term and the pre-factor term are dependent upon $\gamma$. Using the product rule of limits, we analyse the limit of the pre-factor and tracking terms independently and then multiply them together. Using L'Hopital's rule, the pre-factor term in Equation (96) in the limit $B_T \to 0$ becomes

$$\lim_{B_T \to 0} \frac{\tau\kappa\bar{m}T(1-\gamma(B_T))}{2\underline{m}(1-\gamma(B_T)^{T-1})} = \lim_{B_T \to 0} \frac{\tau\kappa\bar{m}T(\frac{B_T}{T})^\alpha}{2\underline{m}(1-(1-(\frac{B_T}{T})^\alpha)^{T-1})} \tag{97}$$

We remove terms that do not depend upon $B_T$ for simplicity and then apply L'Hopital's rule:

$$\lim_{B_T \to 0} \frac{(\frac{B_T}{T})^\alpha}{(1-(1-(\frac{B_T}{T})^\alpha)^{T-1})} = \lim_{B_T \to 0} \frac{1}{(T-1)(1-(\frac{B_T}{T})^\alpha)^{T-2}} \tag{98}$$

$$= \frac{1}{T-1} \tag{99}$$

thus finding the limit of the pre-factor term. As $T > 1$, $\frac{\tau \kappa \bar{m} T}{2 \underline{m}(T-1)} < \frac{\tau \kappa \bar{m}}{\underline{m}}$, we use our analysis from the estimation bound and set $0 < \alpha < \frac{2}{3}$, such that the limit of the tracking term is 0 as expected in stationary scenarios. We can now write the regret bound as

$$\lim_{B_T \to 0} R_T^{\text{off}} < \underbrace{\frac{4\tau\kappa\bar{m}}{\underline{m}}}_{\text{Pre-factor}} \left( \sqrt{\lambda}W + \frac{C_1}{\tau c_{\sigma,\tau}} \sqrt{\frac{d + \log(1/\delta)}{n}} \right)^2. \tag{100}$$

and recover the result of $\mathcal{O}(n^{-1/2})$ in Corollary 3.

### E.4 DETAILS OF APPLYING BERNSTEIN'S INEQUALITY

We restate the norm to investigate:

$$\|\frac{1}{n}\sum_{i=1}^{n} \tau\gamma^{T-1-t_i}\epsilon_i\hat{\phi}_i\|_{(\tilde{\Sigma}+\lambda I)^{-1}}. \tag{101}$$

We then define two vectors $V$ and $Z$, followed by a matrix $M$:

$$V = [\epsilon_1, \ldots, \epsilon_n], \tag{102}$$

$$Z = [\gamma^{T-1-t_1}\hat{\phi}_1, \ldots, \gamma^{T-1-t_n}\hat{\phi}_n], \tag{103}$$

$$M = \frac{1}{n^2}Z(\tilde{\Sigma}+\lambda I)^{-1}Z^{\intercal}. \tag{104}$$

We then express Equation (101) using $V, Z, M$:

$$\|\frac{1}{n}\sum_{i=1}^{n}\tau\gamma^{T-1-t_i}\epsilon_i\hat{\phi}_i\|_{(\tilde{\Sigma}+\lambda I)^{-1}} = \sqrt{\tau^2 V^{\intercal}MV}. \tag{105}$$

We here recall the definition of $\epsilon_i$, which is a 1-sub-Gaussian random variable:

$$\epsilon_i = o_i - \sigma(\tau\langle\hat{\phi}_i, \theta_{t_i}^* - \theta_{\text{ref}}\rangle),$$

$$\mathbb{E}_{o_i \sim p_{t_i}(a_i \succ a_i'|x_i)}[\epsilon_i] = 0, \tag{106}$$

$$\text{Var}_{o_i \sim p_{t_i}(a_i \succ a_i'|x_i)}[\epsilon_i] = \mathbb{E}_{o_i \sim p_{t_i}(a_i \succ a_i'|x_i)}[\epsilon_i^2] - (\mathbb{E}_{o_i \sim p_{t_i}(a_i \succ a_i'|x_i)}[\epsilon_i])^2 \leq 1. \tag{107}$$

As stated in (Hsu et al., 2012), the Bernstein's inequality for sub-Gaussian random variables in quadratic form implies

$$\tau^2 V^{\intercal}MV \leq \tau^2 \left( \text{tr}(M) + 2\sqrt{\text{tr}(M^{\intercal}M)\log(1/\delta)} + 2\|M\|\log(1/\delta) \right)$$

$$\leq \tau^2 \cdot C_1 \cdot \frac{d + \log(1/\delta)}{n}, \tag{108}$$

for some $C_1 > 0$, while $\|M\| = \lambda_{\max}(M)$. Here we used the definition of $\tilde{\Sigma}$ in Equation (17) to show $\tilde{\Sigma} = \frac{1}{n}Z^{\intercal}Z$, and derive for $\lambda > 0$

$$M \prec \frac{1}{n^2}Z(\tilde{\Sigma})^{-1}Z^{\intercal} = \frac{1}{n}I, \tag{109}$$

$$\text{tr}(M) \leq d/n, \tag{110}$$

$$\text{tr}(M^{\intercal}M) \leq d/n^2, \tag{111}$$

$$\|M\| \leq 1/n. \tag{112}$$

