# OpenReview forum: "Right Now, Wrong Then: Non-Stationary Direct Preference Optimization under Preference Drift"
_ICLR.cc/2025/Conference — Submitted to ICLR 2025_

### Official Review · Reviewer_or9J · 2024-11-02

**Soundness:** 3
**Presentation:** 3
**Contribution:** 2
**Rating:** 6
**Confidence:** 4

**Summary:**

This paper introduces Non-Stationary Direct Preference Optimization (NS-DPO), addressing the issue of time-varying preferences in Large Language Models (LLMs). Unlike existing methods, NS-DPO accounts for preference drift by incorporating a Dynamic Bradley-Terry model and exponential weighting in the loss function. Theoretical analysis provides bounds on estimation errors and regret due to non-stationary preferences. Empirical results demonstrate that NS-DPO effectively maintains robustness under preference drift, outperforming baseline algorithms while preserving performance in stationary settings.

**Strengths:**

1. **Method Simplicity and Clarity**: The proposed NS-DPO method is straightforward and easy to follow, making it accessible for implementation.
2. **Theoretical Support**: Rigorous theoretical analysis provides bounds on estimation errors, ensuring the method's reliability.
3. **Robust Experimental Design**: The experiments are well-designed and comprehensive, effectively demonstrating the method's robustness and effectiveness.

**Weaknesses:**

1. **Baseline Considerations**: The paper could benefit from a more comprehensive comparison with alternative methods. Specifically, the most straightforward approach to handling non-stationary preferences would be to re-finetune the model on new data. Adjusting the $\beta$ parameter to control the model update speed or using a retrieval-based system (e.g. RAG) to update memory could also address the issues illustrated in Figure 1. Incorporating these methods as baselines would provide a more robust evaluation of NS-DPO's advantages.

2. **Reward Accuracy Metric**: While reward accuracy is used as the primary evaluation metric, it is important to note that recent study [1] have shown that reward accuracy does not always correlate directly with model performance. Win rate, which measures the proportion of times the model outperforms a baseline, is often a more meaningful metric in practice. Including win rate alongside reward accuracy would provide a more comprehensive assessment of the model's effectiveness.

3. **Model Fine-Tuning Approach**: The paper uses Llama-2-7b-chat-hf as the base model, which is large enough to support full fine-tuning. However, the experiments only explore LoRA fine-tuning. Conducting experiments with full fine-tuning would offer deeper insights into the performance gains and potential limitations of NS-DPO. This could help determine whether the benefits of NS-DPO are consistent across different fine-tuning strategies.

[1] Preference Learning Algorithms Do Not Learn Preference Rankings. CoRR abs/2405.19534 (2024)

**Questions:**

See Weaknesses.

---

> ### Author Response · Authors · 2024-11-22
> **Response to Reviewer or9J**
>
> Thank you for your review and recognising the strengths of our work, these include the simplicity and clarity of our work, the rigorous theoretical analysis ensuring our methods reliability, and the robust experimental design. We provide clarifications and additional context regarding the points you raised.
>
> ## Baseline Considerations
> As clearly addressed in line 245-251 (Assumption 3 of Section 4), our setting assumes **no prior knowledge of when or how preference drift occurs during the data collection process**. Since we cannot identify the onset of "new" preferences in the dataset, ensuring the consistency of preferences is infeasible. Iterative updating of our model regularized to prior versions of the model via $\beta$ is also unfeasible as we consider the offline setting and as such we can only work with the data available and do not collect more data in streaming fashion.
>
> We also disagree with the necessity of RAG as a baseline, for a fair comparison the data within the RAG retrieval database would also vary with time, it is unclear to us then how RAG would address the problem of LLM alignment under temporal preference drift, without a novel method to explicitly do so. As we work in the offline setting it is not an option to collect ‘new’ data for the RAG memory.
>
> ## Win Rate Experiments
> Thank you for pointing out the limitations of reward accuracy as an evaluation metric. We agree that the win rate, which measures the proportion of instances where the model generations outperform those of a baseline, offers a more actionable metric for assessing model generations. However, it is not trivial in our setting to properly evaluate the win rate. Existing works [5] use external LLMs to evaluate the generated responses and it is difficult to prompt LLMs to align with the preferences at the current time step $T$. We are encouraged that the paper the reviewer cited notes that the reward accuracy is a correlated metric with win rate when the trained model remains close to the reference models. Our experiments set $\tau = 0.1$ to ensure this property. If the reviewer has further advice on how to evaluate this metric we would be keen to engage further on this point.
>
> ## Fine-Tuning Strategies
> Thank you again for the suggestion. Based on your suggestion, we have trained additional language models with full fine-tuning for additional experiments. As Figure 6 shows, NS-DPO successfully fine-tunes the entire parameters of the language models, showing that NS-DPO continues to outperform DPO when all the model parameters are fine tuned. Please note that the primary distinction between DPO and NS-DPO lies in introducing the discount parameter γ in the loss function. This design choice should not lead to significant behavioral differences between full model fine-tuning and LoRA fine-tuning. This approach is well-supported in the literature for models of comparable size [1, 2, 3]. We report full fine-tuning results with smaller language models (llama-3.2-1b) due to compute limitations [4].
>
> ## Additional Revisions
> To address concerns regarding the clarity of our paper, we have revised the abstract and the caption for Figure 3. These updates explicitly describe NS-DPO’s behavior under stationary preferences and its relationship to ρdiff. We also modified the experiment section to report full fine-tuning results in Figure 6. All changes have been marked in blue in the updated manuscript.
> We have addressed all the questions the reviewer has raised and updated our manuscript to reflect this. We also reinforced the paper based on the reviewer’s suggestion. We ask that they increase their score or provide further arguments for rejecting our work.
>
> ## References
> [1] Muldrew, William, et al. "Active Preference Learning for Large Language Models." arXiv preprint arXiv:2402.08114 (2024).
>
> [2] Ji, Kaixuan, Jiafan He, and Quanquan Gu. "Reinforcement learning from human feedback with active queries." arXiv preprint arXiv:2402.09401 (2024).
>
> [3] Mehta, Viraj, et al. "Sample Efficient Reinforcement Learning from Human Feedback via Active Exploration." (2023).
>
> [4] Muldrew, William, et al. "Active Preference Learning for Large Language Models." arXiv preprint arXiv:2402.08114 (2024).
>
> [5] Rafael Rafailov, Archit Sharma, Eric Mitchell, Christopher D Manning, Stefano Ermon, and Chelsea Finn. Direct preference optimization: Your language model is secretly a reward model. Advances in neural information processing systems, 36, 2024.

---

> > ### Comment · Reviewer_or9J · 2024-11-23
> >
> > Thank you for your response.
> >
> > `However, it is not trivial in our setting to properly evaluate the win rate.`
> >
> > Could you clarify why employing the win rate as an experimental measure is problematic? Has the author attempted related experiments, or is there a fundamental reason precluding the use of win rate?
> >
> > `Our experiments set $\tau=0.1$ to ensure this property.`
> >
> > Could you explain why a $\tau=0.1$ guarantees that the trained model remains close to the reference models? I suggest that calculating the Kullback-Leibler (KL) divergence might provide a more rigorous measure rather than simply stating that $\tau=0.1$ suffices to ensure this property.
> >
> > In summary, I believe the experiments conducted by the authors are not comprehensive enough, as they rely solely on reward accuracy as the single metric.

---

> > > ### Author Response · Authors · 2024-11-23
> > > **Response to Reviewer or9J**
> > >
> > > We thank the reviewer for their question and for promptly addressing our concerns. We aim to improve our paper by adding additional results and sincerely appreciate the reviewer's feedback in helping us achieve this.
> > >
> > > ## Implementing Win Rate Experiment
> > > In the context of RLHF, the win rate typically refers to the proportion of comparisons or scenarios where a policy or model produces outputs that are preferred by a human evaluator over a competing baseline.
> > >
> > > Regarding the reviewer’s feedback, we would like clarification on the following points:
> > >
> > > 1. Is the reviewer specifically interested in a win-rate comparison against the DPO policy? Additionally, could we align on the mathematical definition of win-rate that the reviewer has in mind? If possible, we kindly ask the reviewer to provide a definition or refer us to a paper where it is explicitly calculated. We are looking forward to providing additional results and making our paper/results more comprehensive. Thank you for helping us.
> > >
> > > 2. Since we are simulating two different human models, it is unclear how to ask/prompt the 3rd-party LLM judge, such as GPT4, to evaluate according to one or the other (assuming the standard win-rate definition).
> > >
> > > We look forward to your guidance and suggestions on these points.
> > >
> > > ## KL divergence of NS-DPO and DPO policies
> > >
> > > Thank you for your comment on providing KL divergence metric to show the deviation of policy parameters from that of reference models. We added in Table 1 the KL divergence values of trained models and reference models (Supervised Fine-Tuned models). NS-DPO policies consistently show lower KL divergences compared to stationary DPO models. The details of computing KL divergence is provided in Appendix D.6.

---

> > > > ### Comment · Reviewer_or9J · 2024-11-24
> > > >
> > > > **Is the reviewer specifically interested in a win-rate comparison against the DPO policy?**
> > > >
> > > > From my understanding, NS-DPO is designed to learn under conditions of preference drift, which aligns with real-world scenarios, suggesting that synthetic experiments alone are insufficient. The win rate stands out as a critical metric for assessing performance differences between models, as it reflects the tangible improvements on benchmark tasks.
> > > >
> > > > **Additionally, could we align on the mathematical definition of win-rate that the reviewer has in mind?**
> > > >
> > > > The standard definition of win rate involves comparisons where a policy or model produces outputs that are favored by human evaluators (or GPT4) over those generated by a competing baseline.
> > > >
> > > > **Given that we are simulating two distinct human models, it remains unclear how to instruct a third-party LLM judge, such as GPT-4, to evaluate according to one model or the other, assuming the standard win-rate definition.**
> > > >
> > > > Therefore, I recommend that the authors attempt a performance comparison on well-established benchmarks, such as alpacaEval2 and Arena-Hard. These benchmarks employ GPT-4-turbo and have demonstrated high consistency with human evaluation. Fundamentally, the motivation behind this setting should ultimately serve practical scenarios, emphasizing the necessity of benchmarks.
> > > >
> > > > **KL divergence of NS-DPO and DPO policies**
> > > >
> > > > Thank you for your experiments. I believe that incorporating KL divergence can enhance the validity of this work. I have improved my score to reflect this point.

---

> > > > > ### Author Response · Authors · 2024-11-24
> > > > > **Response to Reviewer or9J**
> > > > >
> > > > > Thank you very much for your engagement in the discussion, detailed feedback, and increased score. We truly appreciate the effort of the reviewer to make our paper better.
> > > > >
> > > > > Regarding AlpacaEval2, we are uncertain about its relevance to our work. We do not anticipate AlpacaEval2 to account for temporal components. We are not proposing an entirely new alignment algorithm, our approach aligns with DPO in the absence of temporal variations, for which AlpacaEval2 results are already established. Alternatively, we could treat our gamma as a hyperparameter in this scenario and examine whether it leads to a significant difference compared to pure DPO on a benchmark. However, we are unsure how this would substantially strengthen our work. That said, we are open to experimenting with this approach and including our findings in the camera-ready version.
> > > > >
> > > > > Our work looks ahead to future scenarios where preference datasets are collected over extended time periods–a type of dataset currently absent in the literature but likely to emerge as the field evolves.
> > > > >
> > > > > It also emphasizes the importance of personalization to adapt to the evolving preferences of current users. Existing alignment benchmarks do not adequately address these dynamic, time-sensitive challenges, which we believe will be crucial for advancing personalized alignment in the future. For instance, this could involve fine-tuning a personalized LLM to accommodate changes in preferences over time, whether for an individual user, a group of users, or a team within a company.
> > > > >
> > > > > Thank you again for your active participation in the discussion.

---

### Official Review · Reviewer_Cn55 · 2024-11-03

**Soundness:** 3
**Presentation:** 4
**Contribution:** 2
**Rating:** 5
**Confidence:** 3

**Summary:**

This paper introduces a new algorithm, NS-DPO (Non-Stationary Direct Preference Optimization), which extends DPO to address the temporal drift of preferences. The authors provide both theoretical and empirical support for this algorithm. The theoretical analysis derives a regret bound for the algorithm’s convergence in the offline setting. Empirically, NS-DPO is tested on both synthetic and LLM experiments under conditions where preference data shifts over time.

**Strengths:**

1. The paper is well-presented, highly readable, and organized effectively.
2. The theoretical analysis is comprehensive, and the resulting bound, expressed in terms of a drift measure, is particularly insightful.

**Weaknesses:**

1. The study lacks strong motivation. Are there any advantages to incorporating older preference data instead of relying solely on the most recent data, which is more relevant for the model? Considering that RLHF uses much less data compared to pre-training or SFT and that binary preference data is relatively inexpensive to collect, would it be more effective to gather fresh data for model training? Or alternatively, could we focus on collecting only the updated opinions to align the model?
2. It appears that all older data is discounted. How would the algorithm handle a mixture of drifting and fixed preference data, especially if some of the fixed data—though collected long ago—still holds true? Would this data be overly discounted? Is there a mechanism to ensure that genuinely stable preferences aren't unfairly penalized over time?

**Questions:**

1. As noted in the weaknesses section, is all older data— even the valuable data—significantly discounted?
2. In the results presented, $\rho_{diff}$ values are above 0.7. This may or may not be true given real-world preference data. What would the results look like for much smaller values of $\rho_{diff}$?

---

> ### Author Response · Authors · 2024-11-19
> **Response to Reviewer Cn55**
>
> Thank you for your review and for highlighting the strengths of our work, including the clarity of presentation and the comprehensiveness of the theoretical analysis.  However, we disagree with the weaknesses identified by the reviewer.
>
> ## Motivation for Using Older Data
>
> “The study lacks strong motivation” - The offline time-varying problem that we solve in this paper has a strong basis and motivation in the related literature and has been studied in bandit settings [1,2], Bayesian optimisation [3,4], and reinforcement learning [5]. We encourage the reviewer to check out papers in our related work which clearly highlight this.
>
> ## The relevance of older data and collecting new data
>
> In this paper we focus on the **offline learning scenario**, where the dataset is pre-collected, and additional data collection is **not possible**. This makes the effective use of all available data critical. The theoretical guarantees we have obtained in Section 4 are based on **no assumptions about when or how preference drift occurs (line 245-251, page 5)**. This complicates the identification of "fresh" or "updated" preferences.
>
> NS-DPO employs an exponential weighting strategy, which downweights but does not discard older data. This approach allows the algorithm to leverage information from potentially valuable older datapoints while prioritizing more recent data. The advantage of this can be seen in Figure 2, where we compare NS-DPO to SW-DPO, an alternative that considers only the most recent data within a sliding window. Even when SW-DPO uses the optimal window size to capture only data with post-drift preferences, NS-DPO achieves faster convergence. This result highlights the benefit of incorporating older datapoints rather than discarding them outright.
>
> ## Cost of Collecting Preference Data
>
> While it is true that preference feedback can be less expensive to collect than scalar rewards, the cost of human annotation remains significant, particularly for RLHF applications. Recent work on active learning techniques to minimize the need for human queries [6, 7, 8], further illustrating the importance of efficient selection of training data. This reinforces the relevance of our setting, where we aim to maximize the utility of pre-collected datasets under non-stationary conditions.
>
> ## Handling Stable Preferences
>
> We analyze how the algorithm handles a mixture of drifting and fixed preference data in Figures 4 and 5. We test NS-DPO for different values of $\rho_\mathrm{diff}$ which controls the degree of fixed and drifting preferences within the data (see Section 5.1.2 lines 398-400 and Appendix D.1). Figure 3 shows that NS-DPO is robust in settings where no preference shift is observed. **We are confused by this point as it is clearly addressed within our paper**. We have added further analysis of fixed preferences in Appendix C where we analyze the gradient of the NS-DPO loss.
>
> ## Planned Revisions
>
> We have updated the **abstract, the caption for Figure 3** to explicitly explain the behavior of NS-DPO under stationary preferences and its relationship to $\rho_\mathrm{diff}$, and highlight the offline nature of our work. All modifications are marked in blue.
> We have addressed all the questions the reviewer has raised and updated our manuscript to reflect this. We ask that the reviewer increase their score or provide further strong arguments for rejecting our work.
>
> ## References
>
> [1] Yoan Russac, Claire Vernade, and Olivier Cappé. Weighted linear bandits for non-stationary environments. Advances in neural information processing systems, 32, 2019.
>
> [2] Louis Faury, Yoan Russac, Marc Abeille, and Clément Calauzenes. Regret bounds for generalized linear bandits under parameter drift. arXiv preprint arXiv:2103.05750, 2021.
>
> [3] Bogunovic, Ilija, Jonathan Scarlett, and Volkan Cevher. "Time-varying Gaussian process bandit optimization." Artificial Intelligence and Statistics. PMLR, 2016.
>
> [4] Brunzema, Paul, Alexander Von Rohr, and Sebastian Trimpe. "On controller tuning with time-varying Bayesian optimization." 2022 IEEE 61st Conference on Decision and Control (CDC). IEEE, 2022.
>
> [5] Qichao Zhang, Dongbin Zhao. Data-based reinforcement learning for nonzero-sum games with unknown drift dynamics. IEEE Transactions on Cybernetics 49.8 (2018): 2874-2885.
>
> [6] Ji, Kaixuan, Jiafan He, and Quanquan Gu. "Reinforcement learning from human feedback with active queries." arXiv preprint arXiv:2402.09401 (2024).
>
> [7] Muldrew, William, et al. "Active Preference Learning for Large Language Models." arXiv preprint arXiv:2402.08114 (2024).
>
> [8] Das, Nirjhar, et al. "Provably sample efficient rlhf via active preference optimization." arXiv preprint arXiv:2402.10500 (2024).

---

> ### Author Response · Authors · 2024-11-24
> **Response to Reviewer Cn55**
>
> Dear Reviewer Cn55,
>
> As the discussion deadline approaches, we kindly ask for your engagement in the discussion. We would greatly appreciate it if you could let us know whether you are satisfied with our responses and are open to reconsidering your score, or if you have additional questions or require further clarification.
>
> Thank you for your time and feedback!

---

> > ### Author Response · Authors · 2024-11-27
> >
> > Dear reviewer,
> >
> > Thank you for taking the time to review our paper, with the week long extension to the ICLR rebuttal period we are looking forward to engaging in further discussion with you over our rebuttal of your initial review. **We have added further experiment results to the manuscript that support the validity of our method** (see Table 1. Appendix D.6), and **additional explanations and clarifications based on your feedback**, these are highlighted in blue.

---

> > > ### Comment · Reviewer_Cn55 · 2024-11-30
> > >
> > > Thank you for your response and additional experiments. However, I am not convinced why NS-DPO performs as well as DPO when preferences show no drift (figure 3) since the loss functions are significantly different.
> > >
> > > My larger issue, as I have stated before is that a single non-trivial discount factor wouldn't be able to accurately weigh a correct old data point (say "Paris is capital of France" > "London is capital of France" at time t=0) and a currently incorrect old data point (say "Pluto is a planet" > "Pluto is not a planet" also at t=0) differently. For this, a more nuanced approach is necessary -- as compared to the naive discounting method proposed by this paper.
> > >
> > > For these reasons, I maintain my score.

---

> > > > ### Author Response · Authors · 2024-11-30
> > > >
> > > > We disagree with the reviewer on both points. Our paper cannot be understood without paying attention to the main theoretical results in Section 4.
> > > >
> > > > We provide an explanation for why NS-DPO performs as well as DPO in Appendix C of paper - the gradient direction of the NS-DPO loss is the same as that of DPO, leading to similar performance in stationary settings. We also address the stationary case of our algorithm in Theorem 3 of our theoretical analysis.
> > > >
> > > > We firmly reject the larger issue the reviewer has raised. A single discount factor is able to handle preference drift as rigorously shown in our main theoretical results. **If the reviewer does not agree, we ask them to provide a theoretically sound argument addressing Theorem 2 in our work**. We cannot operate on a per-example basis, as we are not concerned with **individual preferences** but with the overall drift in data that is captured by the global statistics B_T.
> > > >
> > > > In conclusion, we respectfully disagree with the reviewer’s comments on the basis of the theoretical and empirical evidence we have already provided in the original paper. We request the reviewer to re-evaluate their score or provide stronger arguments.

---

### Official Review · Reviewer_hwjS · 2024-11-03

**Soundness:** 2
**Presentation:** 3
**Contribution:** 2
**Rating:** 3
**Confidence:** 3

**Summary:**

This paper addresses temporal preference drift - a phenomenon where preferences in datasets collected over long periods undergo gradual or sudden changes. They also proposed Non-Stationary Direct Preference Optimization (NS-DPO) algorithm to  handle this preference drift.

**Strengths:**

- Addressing temporal preference drift is an important and interesting problem.
- The paper is well written.
- The work provides both theoretical and experimental results.

**Weaknesses:**

- While temporal preference drift is important, NS-DPO makes a strong assumption: that older data has a higher probability of changed preference labels, regardless of topic. However, this assumption may not hold in reality. For example, preferences for factual topics (like "What is the capital of France?") remain constant over time.

- The experiments are conducted only on synthetic setups.

- What are the benefits of the algorithm compared to this alternative approach: first training the model on the newest data, then using this trained model to generate new preference annotations for older data, and finally training on the combined dataset?

**Questions:**

Please refer to weakness part.

---

> ### Author Response · Authors · 2024-11-19
> **Response to Reviewer hwjS**
>
> We thank the reviewer for recognising the strengths of our work including the importance of the problem and theoretical and empirical contributions our paper makes. However, we disagree with the weaknesses identified by the reviewer.
>
> ## Assumptions on Preference Drift
>
> The reviewer’s claim on our assumption is incorrect. **We do not assume older data points have a higher probability of changed preferences labels**. In our setting, we do not assume any knowledge of which points change preferences and which points do not. Without this information we must discount all points from the past as they are less reliable than more contemporary points. This is key to achieving our strong regret bounds in Theorem 2. Similar assumptions are widely accepted in various settings including bandits, Bayesian optimization, and reinforcement learning [1–6], we direct the reviewer to our related work for a detailed summary of papers that address this setting.
>
> When there is no preference drift in the dataset, Theorem 2 and Appendix E.3 state that we should set $\gamma = 1$, weighting all points equally regardless of their timestamps. As shown in Figure 3, **NS-DPO matches the performance of vanilla DPO in stationary environments even when** $\gamma < 1$. As such NS-DPO is robust in settings with stationary factual data. Figure 4 and 5 address how NS-DPO performs given different ratios of fixed and changing preferences. The parameter $\rho_\mathrm{diff}$ controls this ratio as clearly stated in line 398 - 400.
>
> ## Experimental Design
>
> We disagree with the claim that our experiments are entirely synthetic. We incorporated real-world datasets with modifications to reflect preference changes. For example:
>
> 1) **GlobalOpinionsQA**: We constructed a dataset exhibiting gradual preference shifts between the preferences of citizens from different countries.
>
> 2) **Helpful-Harmless and Ultrafeedback datasets**: We applied preference drift patterns derived from widely used reward models, such as PairRM and ArmoRM.
>
> This **semi-synthetic** approach with real prompts and responses allowed us to control the timing and intensity of preference shifts. This approach enabled detailed comparisons between NS-DPO and baseline methods under various interesting scenarios.
>
> ## Suggested Alternative Approach
>
> The approach suggested by the reviewer has limitations that NS-DPO addresses more effectively:
>
> 1) **Compatibility with the Setting**: The suggested approach by the reviewer requires knowledge of drift timing, which is **incompatible with the setting of our paper**. NS-DPO does not require explicit knowledge of when drift occurs or ends. This makes it well-suited for handling both sudden and gradual drift without additional steps for labeling or retraining.
>
> 2) **Inaccuracy in Labeling Older Data**: Using a model trained solely on the newest data can introduce inaccuracies when labeling older data, especially if the amount of recent data is limited.
>
> 3) **Simplicity and Robustness**: As shown in Figure 3, NS-DPO achieves comparable performance to vanilla DPO in stationary settings, demonstrating its robustness without the added complexity of iterative labeling.
>
> ## Request for Reevaluation
>
> In response to the reviewer’s comments, we have revised the caption of Figure 3 (page 9) to explicitly highlight that this experiment is conducted without preference drift. We have also added further commentary on stationary preferences in Appendix C. All modifications in the revised manuscript are marked in blue for convenience.
>
> We request that the reviewer reconsider their evaluation, as we address all the highlighted weaknesses within the paper, and we have further revised the paper to clarify specific points. We ask that they increase their score or provide further strong arguments for rejecting our work.
>
>
> ## References
>
> [1] Yoan Russac, Claire Vernade, and Olivier Cappé. Weighted linear bandits for non-stationary environments. Advances in neural information processing systems, 32, 2019.
>
> [2] Louis Faury, Yoan Russac, Marc Abeille, and Clément Calauzenes. Regret bounds for generalized linear bandits under parameter drift. arXiv preprint arXiv:2103.05750, 2021.
>
> [3] Brunzema, Paul, Alexander Von Rohr, and Sebastian Trimpe. "On controller tuning with time-varying Bayesian optimization." 2022 IEEE 61st Conference on Decision and Control (CDC). IEEE, 2022.
>
> [4] Zhou, Xingyu, and Ness Shroff. "No-regret algorithms for time-varying bayesian optimization." 2021 55th Annual Conference on Information Sciences and Systems (CISS). IEEE, 2021.
>
> [5] Qichao Zhang, Dongbin Zhao. Data-based reinforcement learning for nonzero-sum games with unknown drift dynamics. IEEE Transactions on Cybernetics 49.8 (2018): 2874-2885.
>
> [6] Vrabie, Draguna, and Frank Lewis. "Neural network approach to continuous-time direct adaptive optimal control for partially unknown nonlinear systems." Neural Networks 22.3 (2009): 237-246.

---

> > ### Author Response · Authors · 2024-11-27
> >
> > Dear reviewer,
> >
> > Thank you for taking the time to review our paper, with the week long extension to the ICLR rebuttal period we are looking forward to engaging in further discussion with you over our rebuttal of your initial review. **We have added further experiment results to the manuscript** that support the validity of our method (see Table 1. Appendix D.6), and **additional explanations and clarifications based on your feedback**, these are highlighted in blue.

---

> ### Comment · Reviewer_hwjS · 2024-11-28
> **Reply for the rebuttal**
>
> Thank you for the detailed reply and additional experiments.
>
> - Regarding the assumption: as mentioned in the rebuttal, "Without this information we must discount all points from the past as they are less reliable than more contemporary points." This actually assumes that older data is less reliable than recent data. However, this assumption does not hold for factual topics (e.g., "What is the capital of France?"), where older labels are just as reliable as recent ones. How does this assumption affect the algorithm's performance?
> - As shown in Section 5.1.2, the non-stationarity, which is the main focus of the paper, is synthetically introduced based on predefined patterns. I believe it would be more convincing to conduct experiments on real-world datasets. Such datasets could be constructed by tracing data with temporal changes, such as president change or knowledge-editing datasets that may exhibit preference changes over time.
> - Additionally, the suggested baseline does not require knowledge of temporal changes. You could simply evaluate it on the "new" data you defined. I still believe this is a worthwhile baseline to include.
>
> For these reasons, I prefer to maintain my current score.

---

> > ### Author Response · Authors · 2024-11-28
> >
> > We thank the reviewer but disagree with the points raised in their response:
> >
> > ### “... where older labels are just as reliable as recent ones. How does this assumption affect the algorithm's performance?”
> >
> > **We have already addressed this question in the paper.** We direct the reviewer to our experiment section where in **Figures 3,4,5** we consider settings with different mixtures of fixed and changing preferences. As clearly stated on lines 398-400 the $\rho_{\text{diff}}$ parameter, that we vary across experiments, changes these mixtures.
> >
> > ### “I believe it would be more convincing to conduct experiments on real-world datasets”
> >
> > We respectfully and firmly disagree with your claim that our experiments are not convincing. The semi-synthetic datasets use prompts and responses from real world datasets. Constructing a full time-varying dataset is far beyond the scope of our theory focused paper and is often the focus of an entire paper, see [1,2]. **Other works in the RLHF literature are justified through semi-synthetic experiments when suitable datasets are not available [3].**
> >
> > ### “Additionally, the suggested baseline does not require knowledge of temporal changes. You could simply evaluate it on the "new" data you defined.”
> >
> > We respectfully disagree with the reviewer. We direct the reviewer to our previous response, “Compatibility with the Setting”. **In order to extract “new” data, we require information of when the preference drift has ended.** The suggested method uses this knowledge, which violates the assumption we make on the environment. The suggested baseline is also far more complex than the simple solution NS-DPO proposes.
> >
> > In conclusion we respectfully disagree with the reviewers points, and ask them to re-evaluate their score, as the points raised above are already addressed in our original manuscript.
> >
> > [1] Yuntao Bai, Andy Jones, Kamal Ndousse, Amanda Askell, Anna Chen, Nova DasSarma, Dawn Drain, Stanislav Fort, Deep Ganguli, Tom Henighan, et al. Training a helpful and harmless assistant with reinforcement learning from human feedback. arXiv preprint arXiv:2204.05862, 2022a.
> >
> > [2] Ganqu Cui, Lifan Yuan, Ning Ding, Guanming Yao, Wei Zhu, Yuan Ni, Guotong Xie, Zhiyuan Liu, and Maosong Sun. Ultrafeedback: Boosting language models with high-quality feedback. arXiv preprint arXiv:2310.01377, 2023.
> >
> > [3] Chakraborty, Souradip and Qiu, Jiahao and Yuan, Hui and Koppel, Alec and Huang, Furong and Manocha, Dinesh and Bedi, Amrit Singh and Wang, Mengdi, MaxMin-RLHF: Towards equitable alignment of large language models with diverse human preferences. arXiv preprint arXiv:2402.08925, 2024 (edited)

---

### Meta-Review · Area_Chair_FrqK · 2024-12-21

**Metareview:**

This paper introduces NS-DPO, an algorithm addressing temporal preference drift in LLM fine-tuning by using exponential weighting in the loss function. Several key points were raised during the thorough discussion between authors and reviewers:

Strengths: 1) The problem is well-motivated. 2) The paper presents a theoretically sound approach with regret bounds. 3) The method is simple and empirically performs well. 4) The presentation is clear and well-organized.

Weaknesses: 1) The reviewers share the concern that experiments are semi-synthetic: real-world data (prompts and responses) and synthetic reward/preference drifts. The reviewers believe that directly testing and evaluating on real-world data would be more convincing. 2) Questions about whether a single discount factor can adequately handle different types of preference changes. IMO, this is partially addressed by the assumption and theoretical result: knowing $B_T$ (variation budget bound) allows the algorithm to use optimal discount factor $\gamma$ which solves the issue. But this might be a concern in practice. 3) Questions about whether older data is less reliable than recent data and should be all discounted and the reliability of older data can vary. IMO, this question is addressed by theoretical results and experiments on varying the portion of changed preference. 4) Question about evaluation metric (reward accuracy vs. win rate), which is addressed in response.

After discussion among reviewers, all reviewers agree to reject the submission. Given the scores and the discussion, I recommend rejection.

**Additional Comments On Reviewer Discussion:**

There are four questions and concerns raised by reviewers as summarized in meta-review. After the rebuttal, the last two questions are resolved, the second question is partially addressed, while there are different opinions regarding the first concern. After discussion, all reviewers agree to reject the submission and the first concern is emphasized by reviewer.

---

### Decision · Program_Chairs · 2025-01-22

Reject